# Global impacts of marine heatwaves on coastal foundation species

Kathryn E. Smith [1] ✉, Margot Aubin[1], Michael T. Burrows [2],
Karen Filbee-Dexter [3,4], Alistair J. Hobday [5], Neil J. Holbrook [6,7],
Nathan G. King[1], Pippa J. Moore[8], Alex Sen Gupta [9], Mads Thomsen [10,11],
Thomas Wernberg [3,4], Edward Wilson[1] & Dan A. Smale [1]

With increasingly intense marine heatwaves affecting nearshore regions, foundation species are coming under increasing stress. To better understand their impacts, we examine responses of critical, habitat-forming foundation species (macroalgae, seagrass, corals) to marine heatwaves in 1322 shallow coastal areas located across 85 marine ecoregions. We find compelling evidence that intense, summer marine heatwaves play a significant role in the decline of foundation species globally. Critically, detrimental effects increase towards species warm-range edges and over time. We also identify several ecoregions where foundation species don't respond to marine heatwaves, suggestive of some resilience to warming events. Cumulative marine heatwave intensity, absolute temperature, and location within a species' range are key factors mediating impacts. Our results suggest many coastal ecosystems are losing foundation species, potentially impacting associated biodiversity, ecological function, and ecosystem services provision. Understanding relationships between marine heatwaves and foundation species offers the potential to predict impacts that are critical for developing management and adaptation approaches.

The vast majority (~90%) of the excess heat arising from anthropogenic climate change has been absorbed by the upper oceans, leading to multidecadal scale warming across most of the globe[1]. As a result, many marine species have shifted their distributions over recent decades, causing widespread changes in the structure and function of marine communities[2] and impacting the ecosystem services they underpin[3]. As a consequence of global warming, climatic extremes such as marine heatwaves (MHWs; discrete and prolonged periods of warm ocean temperature extremes) have also increased in intensity

and duration over the past century[4], a trend which is projected to intensify further in coming decades[5]. Although a range of complex, interacting oceanographic and atmospheric processes drive MHWs[6,7], the underlying fundamental link to human-mediated warming means projected increases in global temperatures will exacerbate future MHW events and their ecosystem impacts[8].

Extreme temperatures associated with MHWs have been shown to impact the performance of a wide range of marine species, from the molecular to the population level, as critical thermal thresholds are

[1]Marine Biological Association of the United Kingdom, Plymouth, UK. [2]Scottish Association for Marine Science, Oban, UK. [3]Oceans Institute and School of Biological Sciences, University of Western Australia, Crawley, WA, Australia. [4]Institute of Marine Research, His, Bergen, Norway. [5]CSIRO Environment, Hobart, TAS, Australia. [6]Institute for Marine and Antarctic Studies, University of Tasmania, Hobart 7001 TAS, Australia. [7]Australian Research Council Centre of Excellence for Climate Extremes, University of Tasmania, Hobart 7001 TAS, Australia. [8]Dove Marine Laboratory, School of Natural and Environmental Sciences, Newcastle University, Newcastle-Upon-Tyne, UK. [9]Climate Change Research Centre, University of New South Wales, Sydney, NSW, Australia. [10]The Marine Ecology Research Group, Centre of Integrative Ecology, School of Biological Sciences, University of Canterbury, Christchurch, New Zealand. [11]Aarhus University, Department of Ecoscience, 4000 Roskilde, Denmark. ✉e-mail: katsmi@mba.ac.uk

exceeded[9–12]. Acute temperature increases associated with MHWs reduce the potential for individuals to respond via mechanisms such as plasticity or relocation[12]. Responses ranging from reduced ecological performance (e.g., growth, photosynthesis) and failed reproduction, to mass mortality events (MMEs) have been reported globally for marine primary producers, invertebrates, fish, birds, and mammals[9,10,13–17] with far-reaching ecological and socioeconomic ramifications[12,18]. In particular, declines in the abundance and health of foundation species such as macroalgae, seagrass, and hard and soft corals can have a disproportionately large impact on the wider community and ecosystem due to their fundamental role in maintaining ecological processes[19,20]. Foundation species are among the most important species in any coastal ecosystem as they define its identity including the biodiversity it supports and the services it provides. The loss of these foundation species can result in ecosystem collapse, leading to widespread shifts in ecological structure and functioning[19,20].

Our understanding of the biological and ecological impacts of MHWs has increased significantly over the past decade as recognition of and research into these events have developed. MHWs are commonly defined as periods of five or more days where sea temperatures are warmer than the 90th percentile based on a 30-year fixed historical climatological baseline for the location and time of year[21,22]. Events can be classified by intensity into moderate, strong, severe, or extreme categories[23]. Event category and other MHW metrics, such as maximum and cumulative intensity, duration, and rate of onset and timing, can be described and explored using this definition, enabling the comparison of MHW metrics across and between events and linked to biological responses[22]. Several MHW characteristics (e.g. duration, cumulative intensity) have been shown to be useful predictors for biological responses in different locations[11,15,17]. However, a coherent understanding of how MHW characteristics mediate the responses of groups of functionally similar, ecologically important foundation species at the global scale is lacking. A loss of foundation species changes the identity of an environment and can lead to severe declines in the ecosystem services (e.g. tourism and fisheries) that human societies depend on[18–20]. Hence, without such knowledge, the inherent spatial and temporal variability of MHWs makes predicting species responses to different event profiles and in differing biogeographic regions problematic. This challenge is further complicated by variability in responses across a given species distribution. For example, populations persisting towards their warm trailing range edge are expected to be more detrimentally impacted in the future as MHW temperatures exceeding critical thermal limits are experienced more frequently[11,12].

Here, we combine an extensively used MHW framework[21,23] with globally distributed ecological observations to assess responses of coastal marine foundation species (macroalgae, seagrass, scleractininan hard corals (hereafter 'hard corals') and soft coral habitat-forming invertebrates, like gorgonians) to MHWs. Using time-series datasets, we examined changes in macroalgae and seagrass abundance, along with observations of bleaching in hard corals and MMEs on Gorgonian soft corals. We focused on more intense MHWs (classified as 'strong' or greater intensity[23] at some point during their lifetime), that occurred during summertime (June to September and December to March for the northern and southern hemispheres, respectively, and all year round within the tropics (23.4°N-23.4°S)). These events were targeted because strong summer MHWs are more likely to elicit a greater response as species will be closer to their thermal maxima. To examine biological responses to MHWs, we collated a total of 2314 observations spanning 85 marine ecoregions defined by Spalding et al.[24]. We focussed on the dominant habitat-forming foundation species at each location, with the exception of hard corals, where species-level bleaching data were rarely available. Where sites were in close geographical proximity (<8 kilometres) and

sharing common environmental features, we averaged the observed response of the dominant foundation species to individual MHW events, resulting in 1322 observations. Using these data, we first applied linear regression to explore if the number of annual negative responses observed changed over time. We next visually explored trends in the level of responses associated with MHWs across foundation species globally and by ecoregion. We then ran two generalised linear models (GLMs; one for macrophytes, one for hard corals and Gorgonian soft corals) to examine the relationships between foundation species responses and key MHW characteristics (mean, maximum, and cumulative intensity, duration, and maximum absolute temperature) at a global level. We included 'marine ecoregion' and 'point in range' (i.e. where a species was located within its geographical range) as predictor variables in the GLMs. Separate GLMs were run for the two groups because the scale of responses varied; bleaching and MMEs were reported as proportions of the population impacted, whereas changes in macrophytes included both losses and gains and consequently were either positive or negative. Ecoregion was identified as a significant variable in both GLMs. Consequently, we ran separate GLMs to explore foundation species responses to MHW characteristics for the 28 ecoregions, where ≥ 10 co-occurring MHW events and biological responses were available. A further two ecoregions were available with ≥10 data points, but analyses were not run on these ecoregions due to the high number of zeros in the data. Zeros represent locations where no response to an MHW was observed.

## Results
### Global impacts of MHWs on foundation species
We found that MHWs have driven major changes in populations of foundation species globally, with the proportion of negative responses relative to total annual observations increasing significantly over time ($p = 0.021$, $R^2 = 0.15$; Fig. 1). While trends in observed responses varied between foundation species, both globally and across marine ecoregions (Figs. 2a–c and 3a–c), impacts were identified in 79 of 85 ecoregions. In the remaining six ecoregions, no responses to MHWs were observed (all regions where no bleaching of hard corals was observed at site level), although for five of these six ecoregions less than three data points were available. Across hard and soft corals, the range of responses varied from 0 to 100% of sampled individuals being impacted in an ecoregion, while seagrass and macroalgae responses ranged from complete loss to, in some cases, considerable increases in abundance.

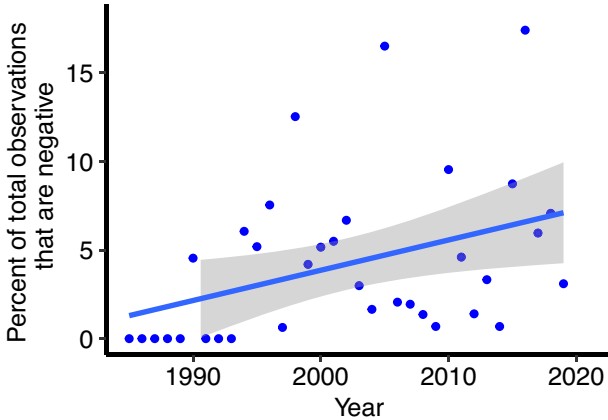

**Fig. 1 | Percent of the total observations that are negative responses to MHWs per year.** The blue trend line indicates the significant relationship between the number of negative responses and year using regression analyses ($p = 0.021$, $R^2 = 0.15$), along with the 95th percentiles around the mean in grey. Source data are provided as a Source Data file.

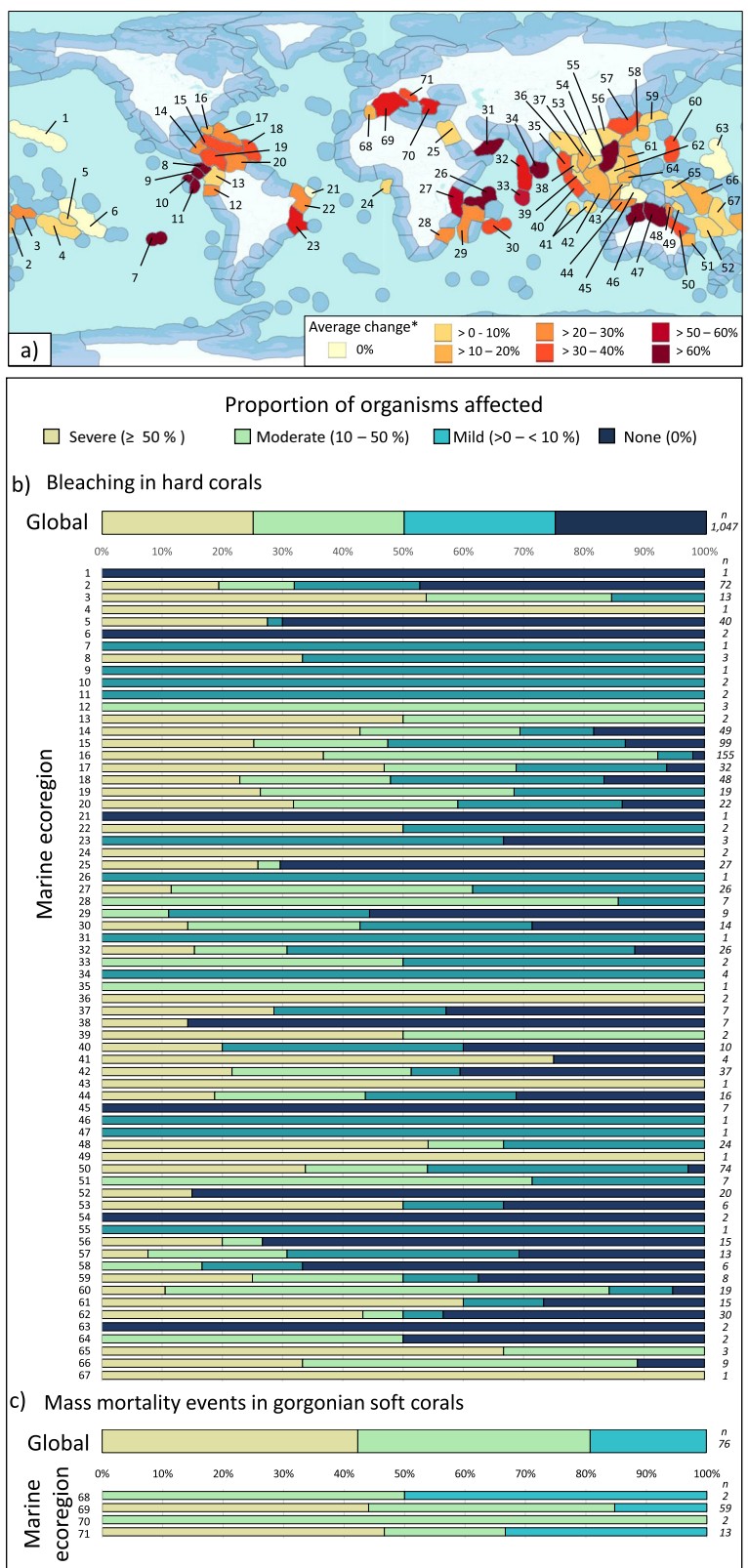

Across ecoregions, where there were enough observations ($n ≥ 3$, 57 ecoregions) to calculate an average change in foundation species, trends in the magnitude of responses varied markedly (Supplementary Data 1). The occurrence of MMEs was greatest in the Western Mediterranean where a single MHW event led to an average of 44.0% mortality across populations of Gorgonian soft corals. For the bleaching of hard corals, the hardest hit ecoregions included South India and Sri Lanka (average 69% bleaching) followed by the Maldives and the East African Coral Coast (51% and 40.3% bleaching, respectively). Comparatively, the least impacted ecoregions included New Caledonia, North and Central Red Sea, Andaman Sea Coral Coast, Cocos-Keeling/Christmas Island, Southern China, and the Banda Sea, where average bleaching remained below 1% across 80 observations gathered following MHW events (Fig. 2a). For seagrass, the greatest

**Fig. 2 | Occurrence of bleaching in Scleractinia hard corals and mass mortality events in Gorgonian soft corals following marine heatwaves by marine ecoregion**[24]. Colour codes represent no change or varying degrees of increased bleaching or mortality. **a** global map of ecoregions included in our analyses. Each ecoregion is coloured by the average observed biological response. Blue coloured ecoregions represent ecoregions where no data were available. The map is adapted from[24] and the relevant shape files available from the Nature Conservancy https://geospatial.tnc.org/datasets/ed2be4cf8b7a451f84fd093c2e7660e3_0/about.
**b**, **c** proportion of responses globally and within each ecoregion that are mild (<10 % of population impacted), moderate (10 – 50% impacted), and severe (≥ 50% impacted). A number of data points identified in each ecoregion are listed on the right. 1 = Hawaii, 2 = Fiji Islands, 3 = Samoa Islands, 4 = Southern Cook/Austral Islands, 5 = Society Islands, 6 = Tuamotus, 7= Easter Island, 8 = Nicoya, 9 = Cocos Islands, 10 = Northern Galapagos Islands, 11 = Eastern Galapagos Islands, 12 = Guayaquil, 13 = Panama Bight, 14 = Western Caribbean, 15 = Greater Antilles, 16 = Floridian, 17 = Bahamian, 18 = Eastern Caribbean, 19 = Southwestern Caribbean, 20 = Southern Caribbean, 21 = Fernando de Naronha and Atoll das Rocas, 22 = Northeastern Brazil, 23 = Eastern Brazil, 24 = Gulf of Guinea Islands, 25 = Northern and Central Red Sea, 26 = Seychelles, 27 = East African Coral Coast, 28 = Delagoa, 29

= Western and Northern Madagascar, 30 = Mascarene Islands, 31 = Western Arabian Sea, 32 = Maldives, 33 = Chagos, 34 = South India and Sri Lanka, 35 = Andaman and Nicobar Islands, 36 = Northern Bay of Bengal, 37 = Gulf of Thailand, 38 = Andaman Sea Coral Coast, 39 = Malacca Strait, 40 = Western Sumatra, 41 = Cocos-Keeling/Christmas Island, 42 = Sunda Shelf/Java Sea, 43 = Northeast Sulawesi, 44 = Lesser Sunda, 45 = Banda Sea, 46 = Bonaparte Coast, 47 = Arnhem Coast to Gulf of Carpentaria, 48 = Torres Strait and Northern Great Barrier Reef, 49 = Southeast Papua New Guinea, 50 = Central and Southern Great Barrier Reef, 51 = Tweed–Moreton, 52 = New Caledonia, 53 = Southern Vietnam, 54 = Gulf of Tonkin, 55 = South China Sea Oceanic Islands, 56 = Southern China, 57 = East China Sea, 58 = South Kuroshio, 59=Central Kuroshio Current, 60 = Mariana Islands, 61 = Eastern Philippines, 62 = Palawan/North Borneo, 63 = Marshall Islands, 64=Sulawesi Sea/Makassar Strait, 65 = Bismarck Sea, 66=Solomon Archipelago, 67 = Vanuatu, 68 = Alboran Sea, 69 = Western Mediterranean, 70 = Aegean Sea, 71 = Adriatic Sea. * Average change is the average amount of bleaching or MME's identified across all locations/events within that ecoregion. Not all ecoregions are averages; only one replicate was available for ecoregions 1, 4, 7, 9, 21, 26, 31, 35, 43, 46, 47, 49, 55, and 67, and thus the colour reflects the single replicate. Source data are provided as a Source Data file.

loss of cover was recorded in the Tweed-Moreton ecoregion (28.6% loss) in Australia whereas only small impacts were observed in the Torres Strait and Northern Great Barrier Reef ecoregion, with changes averaging just 5.7% loss following MHW activity (Fig. 3a). For macroalgae, Northern California recorded the most dramatic declines (39.3% loss of macroalgae density) whereas moderate increases were recorded in the Bassian ecoregion (11.8% gain in macroalgae coverage).

Critically, responses to MHWs were strongly modulated by the location of the surveyed population within species distributions, with the most negative responses occurring towards warm trailing range edges (Fig. 4). Our global-level GLMs indicated point in range to be a statistically significant variable predicting responses to MHWs in both macrophytes ($p = 0.004$) and corals ($p = 0.027$). For macrophytes, sizeable losses were typically recorded in populations persisting towards their warm range edges (20.1% loss), whereas those found towards leading cool range edges exhibited a moderate increase in density or coverage (11.0% gain). MHW-induced bleaching of hard corals and MMEs of Gorgonian soft corals affected larger proportions of the population at warm range edges (23.5%) compared with mid and cool range edges (22.9% and 21.0% bleaching, respectively). Our global-level GLMs also indicated responses in macrophytes to be significantly related to mean MHW intensity ($p = 0.012$), MHW duration ($p = 0.034$), and marine ecoregion ($p = 0.009$). Corals were found to be significantly related to mean and cumulative MHW intensity, maximum absolute temperature, MHW duration (all $p < 0.001$), and marine ecoregion (both $p = 0.001$).

### Increasing MHW intensity and duration exacerbates negative impacts

We identified statistically significant relationships between species responses and key MHW characteristics for 21 of the 28 marine ecoregions where large numbers of observations (> 10) were recorded following strong MHWs (Figs. 5, 6, Table 1). In all ecoregions, foundation species performance declined significantly as one or more MHW characteristics, such as intensity and duration, increased. The most statistically important characteristics varied; for hard corals, maximum absolute temperature was most commonly the strongest predictor of bleaching levels. Conversely, for the other foundation species, the mean intensity was most commonly and significantly linked to responses. Across foundation species types, in many ecoregions, statistically significant relationships were found between biological responses and two or more MHW characteristics (Supplementary Table 2), although for a small number of ecoregions, no significant relationships were evident between differences in responses and any of the MHW characteristics examined. This could be due to factors

including low sample size (e.g. Mascarene Islands) or low overall levels of bleaching in hard corals (e.g. Eastern Philippines). Interestingly, in New Caledonia bleaching in hard corals did not exceed 2.5% despite 20 observations made during MHWs. In the North and Central Red Sea and in Southern China, although commonly above zero, levels of bleaching were consistently low, with a maximum of 12.5% of hard coral bleached during any one event. For Gorgonian soft corals, we identified a significant relationship between MMEs and mean MHW intensity in the Western Mediterranean, whereas in the Adriatic Sea MMEs were significantly linked with MHW duration.

For macroalgae and seagrass, we found that mean and cumulative intensity were the strongest predictors of change in cover, with statistically significant declines in macrophytes observed with increasing MHW intensity (Fig. 6, Supplementary Table 2). Our data also showed that while most responses to MHWs were negative (i.e. a reduction in density or cover of macrophytes), within every ecoregion some positive responses to MHWs were also observed, typically during lower-intensity MHW events. Regardless, observed declines in macroalgae and seagrass cover were mostly associated with increasing MHW intensity.

## Discussion

Marine heatwaves (MHWs) are emerging as pervasive stressors to marine ecosystems, with impacts observed across all trophic levels from primary producers to top predators[9,11,12,25]. Our study presents a globally extensive and coherent analysis of MHW impacts, highlighting the widespread, detrimental effects that these warm ocean extreme events have on a range of coastal foundation species. Given their exceptional importance within marine communities and ecosystems, losses of foundation species will likely have far-reaching implications for local biodiversity, ecological functioning, and the provision of ecosystem services. For example, declines in foundation species lead to loss of biogenic habitat structure, decreased species richness, and shifts in community composition[10,14,26–28], as well as decreased productivity[15]. Considerable ecological and socioeconomic ramifications of foundation species' losses have already been observed globally in response to MHWs[12,14,18,29]. Consequences range from loss of income from tourism or fishing to reduced storm protection and carbon sequestration (Table 1 and references therein). Off Christchurch, New Zealand, large bull kelp has become regionally extinct following an MHW in 2017/2018, replaced by smaller, more ephemeral species[28]. Similarly, in Western Australia, extensive losses of trailing edge macroalgae populations following an extreme MHW in 2011 resulted in an ecosystem-level regime shift from temperate macroalgae forests to warmer-affinity algal turfs that support a more

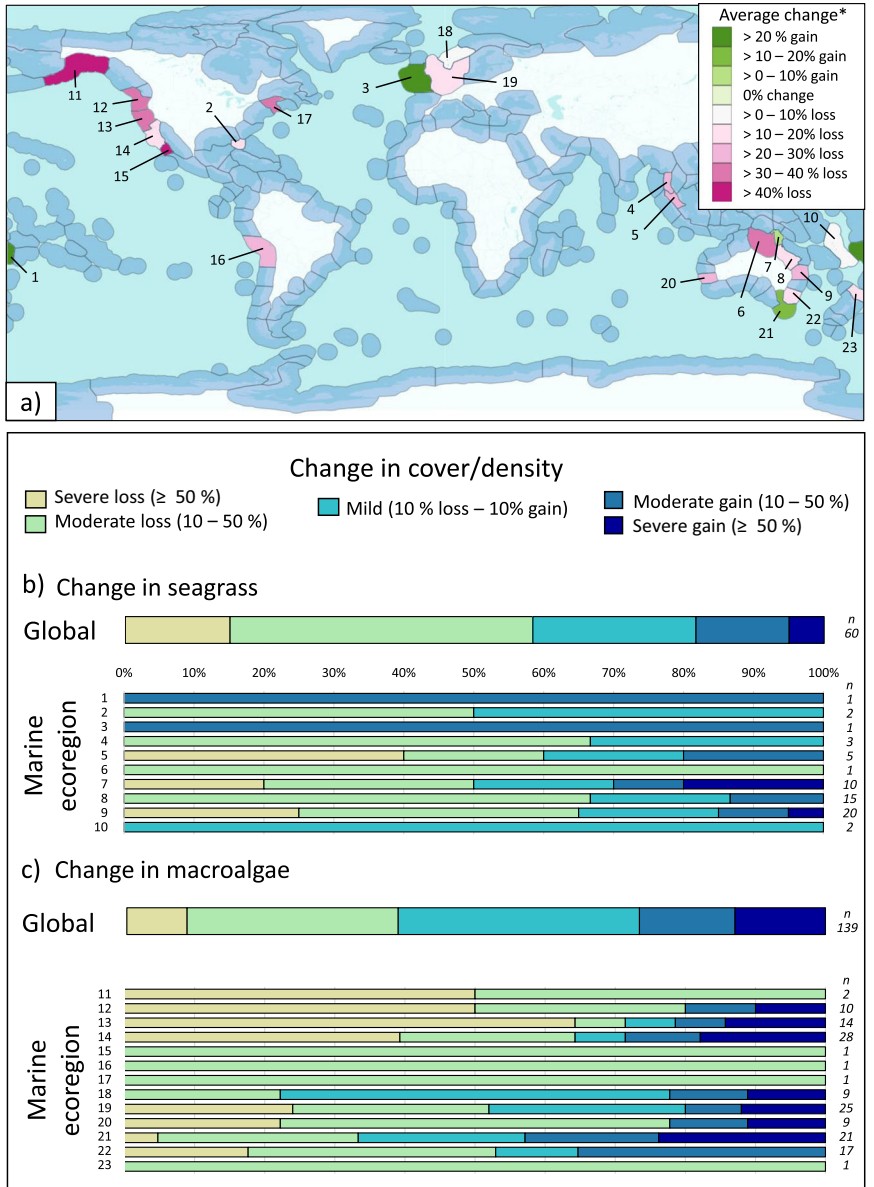

**Fig. 3 | Change in seagrass and macroalgae abundance (percent cover or densities) following marine heatwaves by ecoregion[24]. a** global map of ecoregions included. Each ecoregion is coloured by the average observed biological response. Blue-coloured ecoregions represent ecoregions where no data were available. The map is adapted from[24] and the relevant shape files available from the Nature Conservancy https://geospatial.tnc.org/datasets/ed2be4cf8b7a451f84fd093c2e7660e3_0/about. **b**, **c** proportion of responses globally and within each ecoregion that are mild (10 % gain to 10 % loss), moderate loss (10–50% loss), severe loss (≥ 50% loss), moderate gain (10–50% gain) or severe gain (≥ 50% gain). The number of data points identified in each ecoregion is listed on the right. 1 = Fiji Islands, 2 = Floridian, 3 = Celtic Seas, 4 = Andaman Sea Coral Coast, 5 = Malacca Strait, 6=Arnhem Coast to Gulf of Carpentaria, 7 = Torres Strait and Northern Great Barrier Reef, 8 = Central and Southern Great Barrier Reef, 9 = Tweed–Moreton, 10 = Vanuatu, 11 = Gulf of Alaska, 12 = Oregon, Washington, Vancouver Is., 13 = Northern California, 14 = Southern California Bight, 15 = Magdalena transition, 16 = Humboldtian, 17 = Gulf of Maine, 18 = Southern Norway, 19 = North Sea, 20 = Houtman, 21 = Bassian, 22 = Cape Howe, 23 = Northeastern New Zealand. *Average change is the average amount of bleaching or MME's identified across all locations/events within that ecoregion. Not all ecoregions are averages; only one replicate was available for ecoregions 1, 3, 6, 15–17, and 23, and thus the colour reflects the single replicate. Source data are provided as a Source Data file.

tropicalized community[10], a change that continues to persist more than a decade on from the event[30]. Further north, the same event led to extensive losses of seagrass, resulting in a reduction in commercial fisheries species[29], a decline in charismatic megafauna[14], and the release of significant amounts of carbon from coastal sediments[13].

Our findings provide compelling evidence of the impacts of strong, summer MHWs on the decline in foundation species globally. Foundation species responses were overwhelmingly negative, with the relative number of negative impacts increasing through time, in line with MHW intensification over recent decades[4]. The most detrimental

effects are seen on warm range edge populations, highlighting the potential for MHWs to accelerate range contractions at the equatorward distributions of these species. Our findings provide evidence to support previous research indicating that warm range edge populations, which live closer to their thermal limits, are more negatively impacted by warm water anomalies[11,31]. Further, our research highlights that this pattern is global. While there is evidence to suggest that such events also facilitate an expansion of foundation species at their cool range edge, newly established populations will likely take decades or longer to form productive ecosystems and will support dissimilar

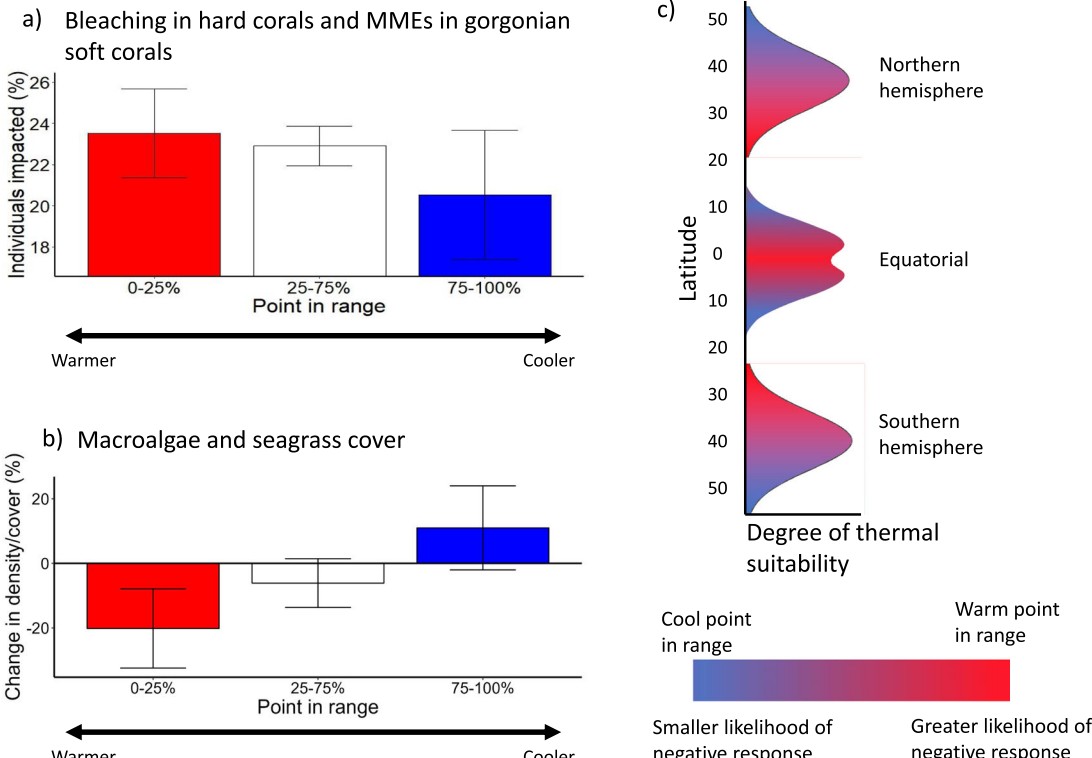

**Fig. 4 | Average responses of plants and animals to marine heatwaves across a species range. a** Percent of populations impacted by bleaching in scleractinian hard corals and mass mortality events in gorgonian soft corals, and **b** Change in macroalgae and seagrass abundance (percent cover or densities). Two-tailed generalised linear models indicate point in range to be a significant factor for predicting responses to MHWs in both corals ($p = 0.027$; $n = 171$, 880 and 73, for 0–25%, 25-75% and 75-100%, respectively) and macrophytes ($p = 0.004$; $n = 34$, 97 and 56, for 0–25%, 25-75% and 75-100%, respectively). Data are presented as mean values +/− SEM. Model parameters are defined in Supplementary Table 1. **c** Illustrative performance curves for species from the northern hemisphere, equatorial region, and southern hemisphere indicating where species are more likely (warm range areas) or less likely (cool range areas) to experience negative impacts of MHWs. Source data are provided as a Source Data file.

communities to the original foundation species with the potential for broad socioeconomic ramifications (Table 1).

At the ecoregion level, we identified clear negative relationships between the heath of foundation species and the severity of MHWs, with foundation species typically exhibiting more detrimental responses as MHW intensity increased. It was, however, interesting to note that despite a general pattern of bleaching in hard corals increasing rapidly at temperatures >30°C, populations in some ecoregions exhibited little to no bleaching following high-magnitude MHWs that exceeded 31°C (e.g. New Caledonia and Southern China). This could be related to local oceanography generating cooler regions that are unresolved by the temperature dataset[32], cloud cover reducing insolation that is necessary for bleaching to occur[33] or indicate a level of thermal acclimation[34], but identification of these regions offers the opportunity for further investigation into the drivers of ecological resilience. Similarly, it was interesting to observe ecoregions where moderate gains were reported (e.g. macroalgae in the Bassian ecoregion). These responses were likely due to the dominant foundational species remaining comfortably within their thermal range throughout the duration of the MHW events, which can lead to no impact or even promote growth and performance. For example, the kelp *Ecklonia radiata* is found at temperatures ranging from 8–25°C[35], but the highest temperature this species was exposed to during MHWs identified during this study in the Bassian ecoregion was -21°C (Supplementary Data 2). As these events fall inside the species thermal range, the higher water temperatures may have enhanced ecological performance. For species located near their cool range edge, these results are perhaps unsurprising; since it is cold temperatures that limit distribution in these locations, it follows that warmer conditions may

promote growth. Certainly, range expansions at the poleward edge of species distributions have previously been reported for kelp and seagrass in the Arctic[36–38] and corals in subtropical regions[39]. Regardless, in many ecoregions, strong MHW events are projected to increase in frequency and duration over the coming decades[5,8], while other ecoregions are characterised by the presence of high numbers of warm range edge species which have a greater likelihood of exhibiting negative responses[11]. It is where these two factors overlap that the most detrimental impacts of MHWs are most likely to be observed[11].

Our analyses add an important contribution to the understanding of MHWs on coastal ecosystems globally. Nevertheless, despite targeting datasets that collect long-term data exploring trends in biodiversity at representative sites (e.g. Reef Check or Seagrass Survey; see Table 2 for details), we recognise elements of the study may be influenced by factors like publication bias or targeted sampling. Similarly, while we targeted strong, summer MHWs because these events are more likely to be stressful and have been associated with previous impacts[11], we acknowledge that our study did not explore responses to MHWs occurring in other seasons. Temperatures experienced during such events will be less likely to exceed upper thermal thresholds and may elicit positive or neutral responses[12], depending on the species and environmental context, and warrant further investigation. Furthermore, we have only assessed direct relationships between MHW characteristics and foundation species responses. Some of the foundation species responses we have observed could be exacerbated by either a compound stressor or may have occurred as an indirect response to MHWs. For example, the loss of seagrass off the coast of Western Australia and the loss of macroalgae off the coast of New Zealand following MHWs were attributed to a combination of

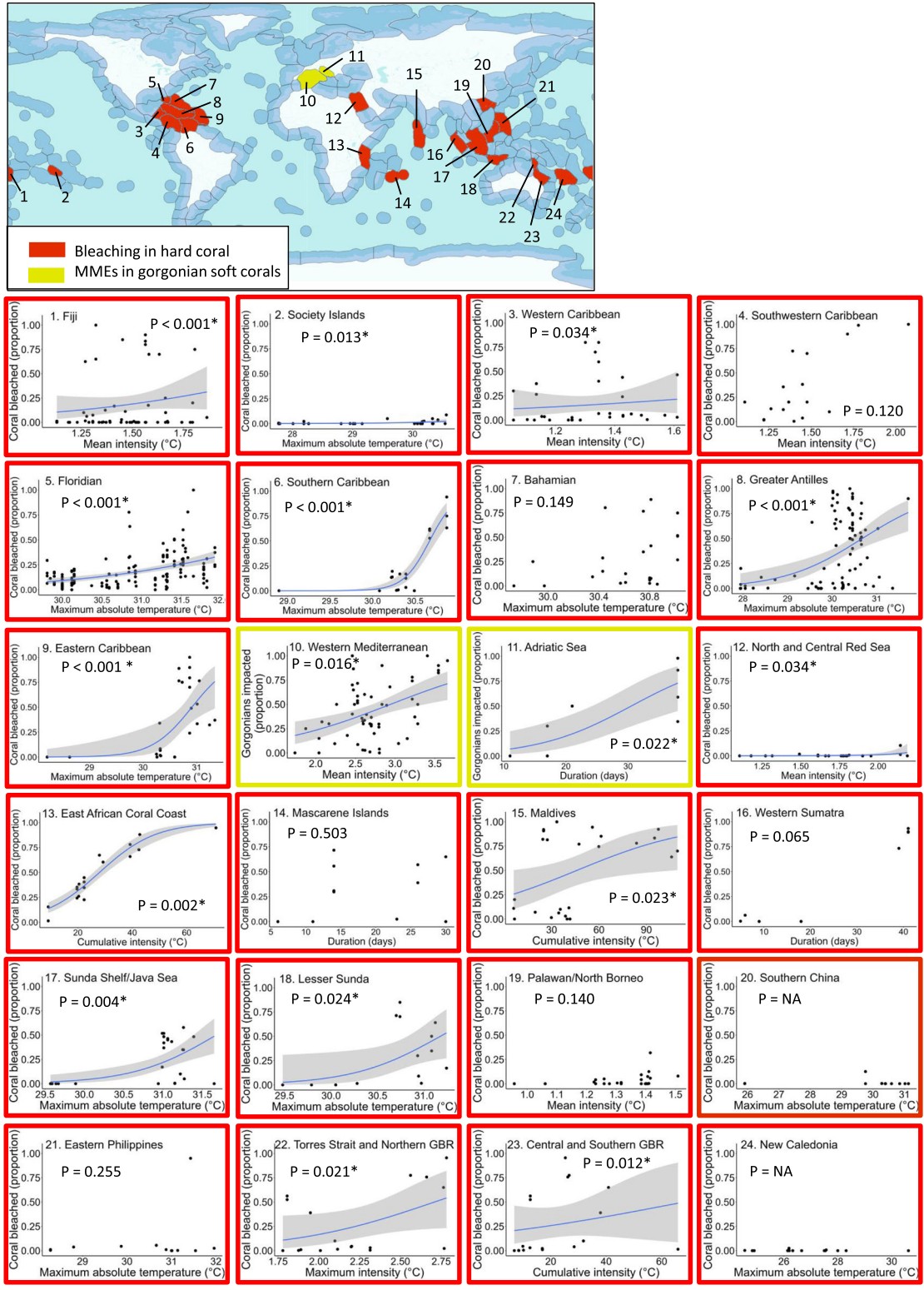

**Fig. 5 | Generalised linear model relationships between foundation species responses and marine heatwave metrics by ecoregion.** For each foundation species group and ecoregion, the MHW metric shown is for the lowest *p* value identified (see Supplementary Table 1 for model parameters and Supplementary Table 2 for all *p* values). * indicates significant *p* values. The colour surrounding each plot indicates the foundation species group. Numbers in the top left corner of each plot refer to the numbered ecoregions on the central map. All GLMs were two-tailed. For significant GLMs, the grey area around the blue trend line indicates 95th percentiles around the mean. GLMs were not run on data from New Caledonia or Southern China due to a high number of zeros present in each dataset. The map is adapted from[24] and the relevant shape files available from the Nature Conservancy https://geospatial.tnc.org/datasets/ed2be4cf8b7a451f84fd093c2e7660e3_0/about. The underlying data for this figure can be found in the figshare database included in the data availability statement.

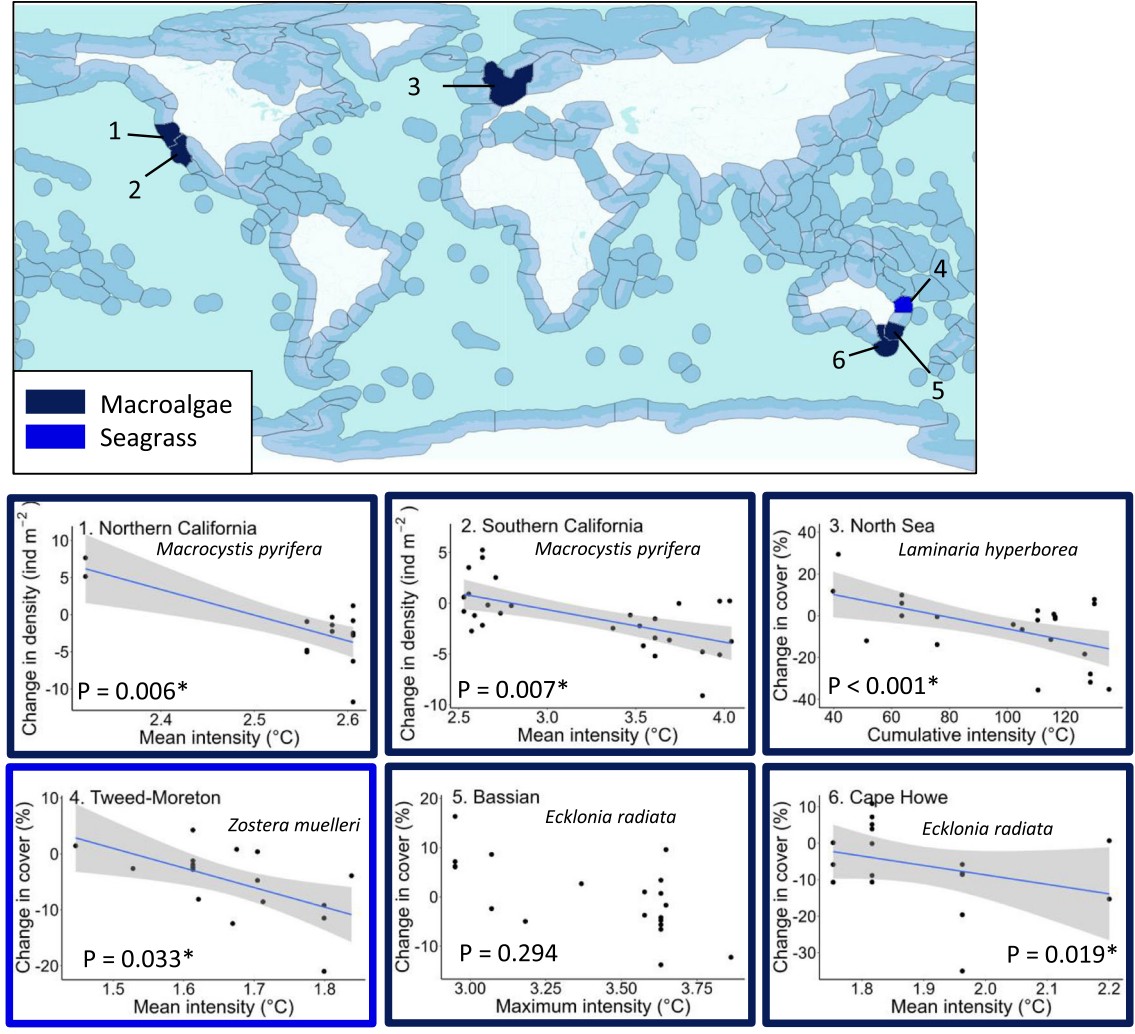

**Fig. 6 | Generalised linear model relationships between foundation species responses and marine heatwave metrics by ecoregion.** For each foundation species group and ecoregion, the MHW metric shown is for the lowest *p* value identified (see Supplementary Table 1 for model parameters and Supplementary Table 2 for all *p* values). * indicates significant *p* values. The colour surrounding each plot indicates the foundation species group. Numbers in the top left corner of each plot refer to the numbered ecoregions on the central map. All GLMs were two-tailed. For significant GLMs, the grey area around the blue trend line indicates 95th percentiles around the mean. The map is adapted from[24] and the relevant shape files available from the Nature Conservancy https://geospatial.tnc.org/datasets/ed2be4cf8b7a451f84fd093c2e7660e3_0/about. The underlying data for this figure can be found in the figshare database included in the data availability statement.

warming, increased sedimentation, and reduced light conditions[14,40]. In a recent global bleaching event, tissue loss and subsequent mortality of hard corals were contributed to by both bleaching and disease induced by warming[41]. Loss of macroalgal forests off the coast of California following 'the Blob' MHW in 2014-2016 was, in some locations, driven by increased herbivory as a consequence of sea star mass mortality (due to wasting disease linked to warming), facilitating urchin population booms[42]. There is clearly a need to better understand the cumulative and secondary responses of a broad suite of species to MHWs, and to identify which species and ecosystems exhibit greater resistance and resilience to extreme warming events.

Identifying relationships between MHW characteristics and biological responses of foundation species across ecoregions takes us a step further towards predicting the impacts of future extreme warming events. Concurrently, such information can be used to ascertain potential climatic refugia or warming-resilient ecosystems by identifying areas at low risk of events or foundation species groups that exhibit some resistance to MHWs[43]. This, coupled with more skillful forecasting[44,45] will improve predictions of how species will respond across their ranges, facilitating management efforts such as ad hoc fishery closures and targeting of alternative species to alleviate the impacts of MHWs in high-risk areas[29,46,47]. In some locations, early management actions have already alleviated the impacts of or even taken advantage of, MHW activity[48]. Building a mechanistic understanding of how other species, or entire communities, respond to MHWs or compound events, including both direct and indirect responses[14,42], will further increase predictive ability. Additionally, gaining an understanding of the recovery of MHW-affected communities will shed light on the longer-term resilience of ecosystems. Further, supporting field observations with laboratory studies[49] or more sophisticated mesocosm approaches[50] will help to identify physiological tipping points of critical foundation species. Ultimately, to mitigate the future impacts of MHWs, it is essential to use a multifaceted approach to understand species-level risks, resilience, and recovery.

## Methods
### Data collection
**Foundation species datasets.** We analysed nine long time-series datasets that reported either density or coverage of macroalgae or

**Table 1 | Ecosystem services impacted by loss of foundation species associated with MHWs**

| Service type | Ecosystem service | Impacts | Foundation Species Example |
|---|---|---|---|
| Provisioning | Raw materials | Reduction in available resources | Loss of kelp forests off Northern California due to MHWs in 2014–2016 impacted commercial kelp harvesting operations and the raw materials (e.g. alginate) this harvesting produces[42] |
| | Food | Loss or shift in the distribution of finfish, shellfish or algae | Record-low recruitment in commercially fished brown tiger prawns in 2012 was attributed to the loss of seagrass during the 2011 MHW off Western Australia[29] |
| | Medicinal resources | Decline in target species | A vast number of medicinal resources come from coral reef communities in places like the Caribbean and Indian Ocean[59]. Loss of corals from MHWs impacts the ability to farm these resources |
| Cultural | Recreation and tourism | Impacted environments are less attractive for recreational and tourism activities | The population decline of abalone due to kelp loss caused by a MHW in 2014–2016 led to closure of the recreational fishery off California. Estimated loss of tourism valued at US$44 million per annum[42] |
| | Spiritual experience | Reduced spiritual experience or availability of spiritual resources | Loss of kelp during a MWH off New Zealand in 2017–2018 impacted the ability of indigenous communities to make sacred 'poha' kelp bags for food storage[63] |
| | Aesthetic appreciation | Non-healthy ecosystems are typically much less aesthetically pleasing than healthy ecosystems | Decline in SCUBA tourism in Southeast Asia in 2010 due to reefs being less attractive because of a coral bleaching event caused by a MHW. Estimated loss of earnings of US$49–74 million[64] |
| Regulating services | Carbon sequestration and storage | Loss of kelp and seagrass impacts blue carbon potential | Loss of >100,000 ha of seagrass in Shark Bay, Western Australia due to a MHW in 2011 led to the potential release of between 2 and 9 Tg $CO_2$ from sedimentary carbon stocks[13] |
| | Extreme events | Reduced natural coastal defences | Loss of coral reefs in the Seychelles due to a MHW in 1998 led to an increase in the wave energy reaching shorelines[65] |
| | Nutrient cycling | Reduced nutrient cycling results in decreased ecosystem health | Reduced nutrient loading in kelp on the eastern and western sides of the North Atlantic has been linked to MHWs[15] |
| | Biological control | Increase in the available niche for non-native species | Loss of Gorgonians during MHWs in the Mediterranean Sea has facilitated the establishment of non-native seaweeds[27] |
| Habitat services | Species | Decrease in biodiversity | Persistent coral bleaching in the Seychelles has led to a long-term shift and reduction in reef fish species and altered assemblage structure[66] |
| | Genetic diversity | Reduced genetic diversity as only warm-tolerant individuals survive | Following a MHW off Western Australia in 2011, genetic diversity in habitat-forming seaweeds was reduced by ~30–65% along 800 km of coastline[67] |
| | Disease | Higher temperatures increase likelihood of disease | The first outbreak of Pacific Oyster Mortality Syndrome in Tasmania, linked to a MHW in 2015–2016 caused significant losses to the Pacific oyster industry[68] |
| | Biogenic structure | Restructuring of entire ecosystems | A MHW in 2011 off Western Australia led to a 100 km range contraction of kelp, which more than a decade later has not recovered[10,30] |

Ecosystem services listed are based on The Economics of Ecosystems and Biodiversity (TEEB) ecosystem services classifications (provisioning services, cultural services, regulating services, and habitat services[69].

seagrass, bleaching in scleractinian hard corals, and mass mortality events (MMEs) in gorgonian soft corals. Datasets with a large spatio-temporal coverage were targeted to maximise the likelihood of identifying marine heatwaves (MHWs) that occurred during data collection across global locations (see Table 2 for details of the datasets used). The datasets used were generally datasets that had been gathered to explore trends in biodiversity at representative sites, rather than surveys which targeted specific impacts or areas. For example, the Santa Barbara Channel Long Term Ecological Research (SBC-LTER) team has been carrying out standardised surveys since the year 2000 on a range of sites, and Reef Check (www.reefcheck.org) has been using trained citizen scientists to survey reefs globally since 1996. We examined the datasets for any overlap (e.g. PISCO[51] data up until 2017 are also available in the dataset from ref. 52) and removed duplicate data points. We also removed data points that were described to have been impacted by stressors other than temperature (for example, Seagrass Watch records all Tropical Cyclones occurring across monitoring periods). Because MHWs are typically identified from sea surface temperature (SST) measurements, we targeted ecological surveys that had been carried out in shallow waters for consistency. We therefore used an arbitrary cut-off of 10 m depth for all datasets, and where multiple surveys were carried out at the same location and time, we used the shallowest records. These criteria resulted in 9027 global

locations with multiple years of data or 25,951 individual data points. All data were reported at the species level, except hard corals that were reported at the order level (Scleractinia).

**Definition and calculation of MHWs.** To identify MHWs from observational SST time series data we used the definition proposed by Hobday et al.[21,23], which defines a MHW as a period of 5 or more days when water temperatures are above a seasonally varying 90th percentile climatological threshold (i.e., consecutive events separated by two or fewer days were analysed as single events). The climatological mean and threshold were calculated over an 11-day window centred on each calendar day using a 30-year fixed baseline from 1983 to 2012[21,23]. The climatological mean and threshold were smoothed using a 30-day running window. MHW intensities were also categorised based on multiples of the difference between the climatological mean and the 90th percentile for any location and time, i.e. intensities between 1-2x are Moderate, 2-3x are Strong, 3-4x are Severe and >4x are Extreme events[23].

To detect MHWs and calculate key metrics we used the R code (https://robwschlegel.github.io/heatwaveR[53]) which follows the Hobday et al.[21,23] MHW definition and intensity categorisation schemes. MHW events were calculated for each location where we had foundation species data, using the 1/4° resolution National Oceanic and

**Table 2 | Foundation species datasets used in the study**

| Dataset name | Foundation species group | Geographic area covered | Years included in this study | Number of available data points pre-processing | Reference |
|---|---|---|---|---|---|
| TRB_KEEN kelp timeseries | Macroalgae | Global | 1982–2014 | 765,762 | 70 |
| IMOS National Reef Monitoring Network* | Macroalgae | Southeast Indian Ocean / Southwest Pacific Ocean | 1992–2021 | 397,026 across relevant sampling methods | www.imos.org.au |
| PISCO kelp forest community surveys | Macroalgae | Northeast Pacific Ocean | 1999–2019 | 266,788 across sampling methods | 51 |
| Dataset from Beas-Luna et al. 2020 | Macroalgae | Northeast Pacific Ocean | 1999–2017 | 998,824 datapoints across ~ 325 species | 52 |
| LTER Santa Barbara Channel | Macroalgae | Northeast Pacific Ocean | 2000–2021 | 1046,718 across relevant sampling methods | 71 |
| Seagrass-watch Global Seagrass Observation Network | Seagrass | Global | 2005–2022 | 5914 assessments across multiple species | www.seagrasswatch.org |
| Global Coral Bleaching Database | Scleractinia hard corals | Tropics | 1980–2020 | 34,846 reports of reef state | 54 |
| Reef Check† | Scleractinia hard sorals | Tropics | 1997–2020 | 15,626 assessments of multiple species | www.reefcheck.org |
| Dataset from Garrabou et al.[17] | Gorgonian soft corals | Mediterranean Sea | 2015–2019 | 985 | 17 |
| T-MEDNet | Gorgonian soft corals | Mediterranean Sea | 1985–2020 | 716 | www.t-mednet.org |

*Data were sourced from Australia's Integrated Marine Observing System (IMOS) – IMOS is enabled by the National Collaborative Research Infrastructure Strategy (NCRIS). It is operated by a consortium of institutions as an unincorporated joint venture, with the University of Tasmania as Lead Agent. †All reef check data is incorporated into the Global Coral Bleaching Database and so was not explored independently.

Atmospheric Administration (NOAA) Optimum Interpolation SST V2 data. Analysed MHW metrics included mean intensity (the average SST anomaly across the event), maximum intensity (the maximum SST anomaly reached during an event), cumulative intensity (the sum of the daily intensities across the event), maximum absolute temperature (the highest recorded SST during an event) and duration (the number of days an event lasts). All events categorised as Moderate were removed, leaving only Strong, Severe, and Extreme category events. Higher magnitude events were targeted because a) MHWs of a magnitude of Moderate intensity have been present throughout the satellite record making it difficult to differentiate between events whereas events of Strong or greater intensity occur less frequently[8,12,17] and b) most key events which have elicited recorded ecological responses have been category Strong or greater[54]. For all locations outside the Tropics of Cancer and Capricorn (23.44° N to 23.44° S), the remaining events occurring outside 'summer' were eliminated. For the purposes of this study, we defined boreal summer as 1st June–30th September and austral summer 1st December – 31st March, following Smale et al.[11]. Summer events were targeted because species are naturally closer to their thermal maxima during this season, increasing the likelihood of a response being observed[55].

**Foundation species responses to MHWs.** To identify biological responses, we cross-referenced detected MHWs with foundation species data to identify periods where Strong or above-category MHWs fitting the above criteria overlapped with the biological datasets. If events occurred during multiple years at any one site, biological data were only included if at least two years had passed since the previous MHW categorised as Strong or greater intensity, offering foundation species the opportunity for some recovery[56–59]. We focussed on the dominant habitat-forming species at each location, with the exception of hard corals, where species-level bleaching data were rarely available, and followed two methodologies depending on the target foundation species.

1. For macroalgae and seagrass (in combination referred to as 'macrophytes') we used time-series data to compare densities (counts per area) or percent coverage from directly before, but not during, a strong MHW event to 6–12 months after that MHW event. This post-MHW duration was used because there is typically a lag between MHWs and changes in plant density or cover. In each location, we focussed on the dominant species only (e.g. *Macrocystis pyrifera* off California). Where possible we averaged measurements of macrophytes pre-MHW over 2–3 years to account for natural variability. Data gathered at the same time of year were used to compare pre- and post-MHW conditions. Density/coverage from pre-MHW was subtracted from the same data post-MHW to determine if macrophytes had increased or decreased, and to what extent.

2. For bleaching in hard corals and MMEs in Gorgonian soft corals, we examined MHW events that occurred at most 16 weeks prior to ecological surveys. Bleaching and MMEs in these foundation species groups typically occur during or soon after MHWs[60,61], and the shorter period prior to surveys was chosen to reflect this.

To calculate the relevant MHW metrics for each biological response, consecutive events that fell within the timeframes being examined were combined. Where any event ran over the confines of the timeframes, the event metrics were trimmed to include only intensities that had occurred within the defined period, i.e. 'summer', or 16 weeks prior to surveys. Following the above criteria, we compiled a total data set comprising 2314 sites and time periods. Data from sites that were geographically similar were then averaged for each MHW event; this was defined here as sites within 8 kilometres of each other sharing common environmental features (adapted from Garrabou et al.[17]). The resulting localized areas were then reclassified into marine

ecoregions[24] resulting in 1322 observations across 85 ecoregions (139 for macroalgae, 60 for seagrass, 1047 for hard corals and 76 for gorgonian soft corals). Ranges of the corresponding MHW characteristics for ecoregion are listed in Supplementary Data 2.

## Data exploration and analysis

**Trends in MHW responses across time.** We used our initial 25,951 site/year data points along with our 2208 sites and time periods that were impacted by a strong, summer MHW, to assess how negative responses to MHWs have changed over time. All observations were included here rather than the responses averaged within 8 km, to ensure the data was comparable with the 25,951 initial data points, which had not been averaged. We calculated the number of initial data points that had been gathered in any one calendar year along with the number of negative responses from our 2314 data points occurring within a calendar year (excluding either neutral or positive responses). We then calculated the percentage of annual data points that showed a negative response to MHWs. We applied regression analysis on the resulting data to assess if the proportion of sites examined that were negatively impacted by MHWs changed over time.

**Global and ecoregion-level trends in responses.** We used the 1322 biological observations we had gathered to visually explore trends in global and ecoregion-level responses to MHWs across foundation species. For each foundation species group, we determined the proportion of observations that fell into different effect sizes, first on a global scale, and then by marine ecoregion. The effect sizes comprised mild (<10% of the population impacted), moderate (10–50%), and severe (>50% of the population impacted) inhibition (loss) or facilitation (gain). For bleaching in hard corals and MMEs in gorgonian soft corals only inhibition was recorded, whereas for macroalgae and seagrass, both inhibition and facilitation were recorded. For all ecoregions with a minimum of three data points we also averaged all responses for each foundation species group independently to explore differences in global responses. Where data on bleaching in hard corals were only available as ranges (mild (1–10%), moderate (11–50%) or severe (≥50%)), the range was replaced with conservative values of 1%, 11%, and 51%, respectively, to enable data to be incorporated.

We next combined our data across all foundation species for either all the foundational macrophytes or corals, into two separate analyses to statistically explore the relationships between independent variables (MHW characteristics, ecoregion, species point in range) and foundation species responses. Species points in range were assessed as follows: for all species-level data, the full latitudinal ranges were determined from the Ocean Biodiversity Information System (OBIS; https://obis.org) following Smale et al.[11]. For hard corals, where species-level data were not available, we gathered data from OBIS for the full latitudinal ranges of common reef-forming coral genera (*Acropora, Pocillopora, Porites, Favites* and *Goniastrea*; 35.03°N–35.06°S). For this group, which spans the equator, we considered the trailing range edge as the equator and leading range edges as the poleward extents, with a point in the range being expressed as the distance between the equator and northern or southern extent (see Fig. 4c for additional information on points in range). Data were analysed with Generalised Linear Models (GLMs) weighted by the number of sites that were averaged within a geographically similar localised area, as described above. For macroalgae and seagrass direct increases or decreases in counts or percent cover were modelled, so Gaussian error structure was used. For bleaching in hard corals and MME in Gorgonian soft corals, data were expressed as proportions of the population, so a Quasibinomial error structure was used. For each dataset, the Variance Inflation Factor (VIF) was calculated to account for collinearity in independent variables. The largest VIFs were removed sequentially until all remaining independent variables had a VIF < 10. For GLMs run using Gaussian error structure, the Akaike

Information Criterion (AIC) score was also examined to help determine the best fitting models. Model residuals were examined, and outliers were removed if necessary (one for corals and three for macrophytes).

Ecoregion was identified as a significant variable affecting responses in both macrophytes and corals. Consequently, for all ecoregions with ≥ 10 datapoints, we ran additional GLMs to test for the relationships between MHW characteristics and foundation species responses at the ecoregion level. Here, our data were reduced to 1133 observations (128 for macroalgae, 60 for seagrass, 869 for hard corals, and 76 for gorgonian soft corals). An additional 189 observations from across these 30 ecoregions were not included in these analyses either because species were only reported to genus level (for 11 macrophyte locations), or because the responses reported within a range (i.e., bleaching in hard corals reported as mild, moderate, or severe, for 178 hard coral locations) were deemed too coarse for these analyses. For macroalgae and seagrass, we compared responses by a single dominant primary foundation species for each ecoregion (i.e., *Ecklonia radiata* for Bassian and Cape Howe ecoregions, *Macrocystis pyrifera* for Northern and Southern California ecoregions, *Laminaria hyperborea* for the North Sea ecoregion, and *Zostera muelleri* for the Tweed-Moreton ecoregion). No analyses were run for seagrass in the Central and Southern Great Barrier Reef ecoregion or for macroalgae in the Oregon, Washington, Vancouver Coast, and Shelf ecoregion, because data for ≥ 10 observations for a single, known species was not available. For hard corals and gorgonian soft corals, we compared responses across all observations within an ecoregion. No analyses were undertaken for the Samoa Islands, East China Sea, or Mariana Islands because ≥ 10 accurate data points (i.e. presented as a number instead of a range) were not available for these ecoregions (see Supplementary Data 1). GLMs were again weighted by a number of sites within a localised area for each ecoregion or group of ecoregions. As previously, the Gaussian error structure was used for macrophytes, and Quasibinomial error structure was used for corals. Again, the largest VIFs were removed sequentially until all remaining independent variables had a VIF < 10, and for GLMs run using Gaussian error structure the AIC score was examined to help determine the best fitting models (Supplementary Table 1). For each dataset, model residuals were examined and outliers were removed if necessary. All GLMs were conducted in RStudio v. 2022.12.0+353[62].

**Potential sources of non-independence.** We acknowledge that non-independence of data is a potential issue in our dataset and we have corrected this to the best of our ability. To avoid spatial autocorrelation we averaged responses from sites that were geographically similar and in close proximity. We also recognise that we have used multiple sites from the same dataset. However, each site was typically surveyed by a different observer and therefore we elected to retain the full dataset. Finally, we recognise that similar species within a broader clade may have similar responses. However, our dataset shows large variation in responses for a single species across different studies and we therefore again elected to retain the full dataset, to ensure we had the highest power to detect the influence of MHW characteristics.

## Reporting summary

Further information on research design is available in the Nature Portfolio Reporting Summary linked to this article.

# Data availability

The datasets used in this study are shown in Table 2. The processed data used in this study are available in the following figshare database: https://figshare.com/s/b7a4f926c746b2b9cc3a. Sea Surface Temperature data (dataset: ncdcOisst21Agg_LonPM180)used to determine the presence of marine heatwaves was downloaded from https://coastwatch.pfeg.noaa.gov/erddap/. Source data are provided with this paper for Figs. 1–4. The data underlying Figs. 5 and 6 can be found

in the above figshare database. All datasets used in this study are freely available and the conditions of access were followed for each. Source data are provided in this paper.

## Code availability

All codes required to identify marine heatwaves are freely available at https://robwschlegel.github.io/heatwaveR[62].

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

## Acknowledgements

We thank the Australian Integrated Marine Observing System (IMOS), which is enabled by the National Collaborative Research Infrastructure Strategy (NCRIS) for providing data on macroalgal cover in the Southeast Indian Ocean and Southwest Pacific Ocean. Data on mass mortality events were extracted from MME-T-MEDNet database on mass mortality events in the Mediterranean (https://t-mednet.org/MME). Data on seagrass were extracted from https://www.seagrasswatch.org. Data from Reef Check are available at https://www.reefcheck.org. We thank the contributors to Seagrass Watch, T-MEDNet, Reef Check, Sabah Biodiversity Centre, and Reef Check Malaysia. D.A.S. was supported by a UK Research and Innovation Future Leaders Fellowship (Grant MR/X023214/1). T.W. was supported by the Australian Research Council (grant DP200100201). M.S.T. was supported by a University of Canterbury Seeding Grant and the New Zealand Ministry of Business, Innovation, and Employment (Toka ākau toitu Kaitiakitanga – building a sustainable future for coastal reef ecosystems). P.J.M. and M.T.B. were supported by the Natural Environment Research Council Newton Fund (Grant NE/S011692/2). N.J.H. was supported by the ARC Centre of Excellence for Climate Extremes (CE170100023) and the National Environmental Science Program Climate Systems Hub (Project 2.10). ASG was supported by an Australian Research Council Future Fellowship (FT220100475). We acknowledge the Marine Heatwaves International Working Group (https://www.marineheatwaves.org) and several of its workshops for fostering much of this discussion.

## Author contributions

K.E.S., M.T.B., A.J.H., N.J.H., P.J.M., A.S.G., M.T., T.W. and D.A.S. conceived the study. K.E.S. and D.A.S. designed the analyses. K.E.S., M.A., N.G.K. and E.W. collected the data and conducted the analyses. M.T.B., K.F.D., A.J.H., N.J.H., P.J.M., A.S.G., M.T., T.W. and D.A.S. provided technical support and conceptual advice. K.E.S. wrote the paper with contributions from all authors.

## Competing interests

The authors declare no competing interests.
