## [Peer Review File · Nature Communications]

Global impacts of marine heatwaves on coastal foundation speciesREVIEWER COMMENTS

Reviewer #1 (Remarks to the Author):

Some previous studies have provided evidence that MHWs are impacting foundation species (including corals, kelps and macroalgae) across different locations and regions (e.g., Mediterranean Sea, Caribbean Sea, Indo-Pacific...). However, to my knowledge, the present work by Smith et al. is the first study providing a comprehensive, quantitative analysis of the impact of MHWs on critical foundation species across the globe, as well as evidence on how MHWs characteristics (e.g., average intensity, maximum temperature...) and geographic factors (e.g., location within species ranges) may modulate responses. Specifically, the authors combine time series of empirical ecological data (i.e., changes in abundance for macroalgae and seagrasses, and bleaching or mortality observations for corals) and a MHW framework, to quantitatively examine responses of key, habitat-forming foundation species (macroalgae, seagrass and corals) to MHWs in 778 coastal areas located across 72 marine ecoregions. The authors found compelling evidence that intense, summer MHWs play a significant role in foundation species decline. The authors also found that MHW intensity, absolute temperature and location within a species' range are critical factors mediating impacts. Moreover, foundation species have not been affected in some ecoregions, suggestive of some resilience. Overall, I consider that the study is highly timely, addresses a highly relevant topic that is of global interest and has critical and novel results that not only support the conclusions and claims, but also expand our previous knowledge on the topic. The consequences of the reported global decline in marine foundation species driven by MHWs are undeniable for both ecosystems and human societies. The paper is also very well written, with nice figures and sound statistical methods that will be reproducible upon publication of the data.

Nevertheless, I have some comments that the authors may need to address and/or clarify, which I attach below:

#Title and abstract

Not totally necessary...but since the paper is only focused on observations shallower than

10 m (i.e., cut-off of 10m depth for all datasets), it may be a good idea to include the word “shallow” in the title and/or in the abstract so the readers can easily realize that the paper focuses on this zone. I say this because the impacts of marine heatwaves at deeper depths may be different in different regions even in coastal areas, depending on different factors such as for instance the presence/absence of thermocline.

#Main text

Lines 44-117 and 119-187: I am sure authors were planning to do it in the future, but just in case, it would be great to see the headings for these sections as Introduction and Results, respectively.

Lines 122-124, Figures 1-2: Maybe it is just my curiosity, but; would it be possible, by simply looking at the text or at Figure 1, to know which 6 ecoregions are the ones that are not impacted ? It seems to me that those regions with little impact (<10% of the population impacted) and those with no bleaching at all (0%) have been pooled together as (0-10% category). For instance, when looking at the Extended Data Table 1, I can see that Banda Sea was not impacted. However, Banda Sea appears as 0-10% yellow color in the map (panel a), or mild impact (panel b). While the legend is up front with it and says 0-10% (which includes 0), I believe it would be better to have a separate category for those non-impacted ecoregions (0%; maybe white colors in the map?), and another one for 1-10% (yellow). Otherwise, it may create the feeling of mild impact in a region that is not impacted at all when looking solely at the map. Also, without doing such separation, it is not possible to know which are those ecoregions that are not impacted and have only one data point, since the only way of detecting the true zero is to go to Extended Data Table 1 to see the average, and the average is not shown for regions with less than 3 points.

Another point that I believe should be addressed is the legend capture with color codes of Figures 1 and 2, panel a. It says “average response”. However, there are some regions whose values (and colors) are not averages but results of a single data point. Authors are upfront with it mentioning the number of replicates for each region in panels b and c, as well as in the main text and in Extended Data Table 1. However, as it is now, the panel a by

itself is not totally correct since it still says average responses. Maybe authors could add asterisks in those ecoregion numbers linked in the map to regions that could not be averaged (such as 25, 26 in figure 1 panel a) and then explain what those asterisks mean in the figure text. Alternatively, authors may prefer to add an asterisk in the head of the legend (average response*) so then they can explain that not all ecoregions of the map are averages, or any other solution...

Another concern in Figures 1 and 2 is related to how categories have been divided in panel a. If 0-10 and 10-20 are different categories; what would happen if there was an average of exactly 10, or 20? Would they be assigned to the first 0-10 or to the second 10-20 group? Shouldn't the categories be divided such as 0, 1-10, 11-20, 21-30...etc to avoid that issue? It has been done in panels b and c and to me it is clearer...

Finally, since 'other' habitat-forming invertebrates refers here (in Figure 1) to gorgonians, and gorgonians are also corals (octocorals), the header of panel c should not say non-coral habitat forming invertebrates, but something like soft coral habitat-forming invertebrates, or non-hard coral habitat forming invertebrates.

Line 129-142 (and Extended Table 1): I believe there is a small issue here. For corals, the values in these lines are in %0-100 (e.g., 44% in Mediterranean), as shown also in Extended Table S1. However, for macrophytes, while values are also %0-100 in the text (e.g., 39.3% loss in North California), the corresponding values are extremely low (0.4 loss in California) in Extended Table 1 (while they were supposed to be average response %0-100 as well). I believe that the data may have mistakenly been written as proportions 0-1 in Extended Table 1 for macrophytes, instead of percentages 0-100 (as it was for corals).

Line 169: Is the reference of Table 1 actually referring to Extended Table S2?

Line 176: Since other habitat-forming invertebrates refers to gorgonians, consider to specify it here (and maybe throughout the text) (i.e., gorgonians)

#Discussion

Line 222: It would be great to add here some examples of the regions that did not see bleaching despite registered MHWs.

Line 248: The authors mention how analyses such as the one performed in the study can be used to ascertain potential climatic refugia or warming-resilient ecosystems by identifying areas at low risk of events or foundation species groups that exhibit some resistance to MHW. Authors also show in the results how some few ecoregions and systems were not impacted or even gained cover, suggesting some potential resilience. Authors do point at some potential causes (oceanography, cloud cover or indicate a level of thermal acclimation...) of the observed lack of bleaching, but I am maybe missing a little bit more discussion about whether the identified non-impacted regions in this study could (or not) actually be potential climatic refugia...and why authors believe so. Similarly, I miss a little bit more information about why some seagrasses and macroalgae have avoided impacts (Torres Strait and Northern Great Barrier Reef) or even increased cover (Bassian) following sometimes severe MHWs.

#Methods

Line 269: If possible, I would rather see Extended Data Table 3 as a Table in the main manuscript, since it contains information that is highly relevant.

Line 335: Please, either use the word gorgonian, or specify that other habitat-forming species refer to gorgonians only

Line 358-360: but single points also appear in the maps of Figures 1 or 2 as averages.

Line 382 (and line 115): I am a little bit confused here. It says: "for all ecoregions with ≥ 10 datapoints we ran additional GLMs to test for the relationships between MHW characteristics and foundation species responses at the ecoregion level." However, ecoregions with exactly 10 datapoints were excluded in Figure 5 (and Extended Table S2) with only one exception (Lesser Sunda). Similarly, for Figure 4 (and Extended Table S2), only ecoregions with >10 datapoints were used (those with 10 were excluded as well), and

Central and Southern Great Barrier Reef (with 15 points) was excluded. It doesn't seem to be consistent; am I missing something?

*Adding to my previous comment, in Line 161, it says that only ecoregions with > 10 points were used, which makes me believe that the \geq symbol used in lines 382 and 115 was wrong. That would explain most of it. However, I still do not see why Lesser Sunda (with 10 points) was the unique 10-point region used for corals, nor why Central and Southern Great Barrier Reef (with 15 points) was excluded for the macrophytes case. Maybe It was explained somewhere and I didn't see it, but please clarify.

#General comment for Figures 1 and 2

I would prefer a clearer separation between panel a and panels b-c in both figures, since as it is now, the color codes of panels b,c (located just below the map) make the impression of being part of panel a. Other option could be to put the color legend of panels b/c below these panels, instead of in below panel a.

#Extended Data

Extended data table 1:

- It would be nice to see also de SD, minimum and maximum values for each ecoregion
- According to this table, Bassian and Southern Norway have the same average response (%) = 0.1 gain. However, in Figure 2, panel a, whereas Southern Norway has an average response of 0-10% gain, Bassian region seems to have a color of 10-20% gain. If data in Extended Table 1 were correct for macrophytes (0.1 and 0.1), these values should also match in the map and correspond to the 0-10% interval. However, as mentioned above, I believe that the data may have mistakenly been written as proportions 0-1 in the table S1 (instead of percentages 0-100). Since they have also been rounded, they now look equal (0.1 and 0.1), when they should be 11.8% for the Bassian region, and something below 10% for the Norway one? Please, clarify.

Dr. Daniel Gómez-Gras

Reviewer #2 (Remarks to the Author):

What are the noteworthy results?

This manuscript assesses the response of coastal foundation species to marine heatwave metrics, including max absolute temperature, mean/max/cumulative intensity, and duration. The novel part of this analysis is that the authors examine corals, seagrasses, and macroalgae in the world's oceans. They find many ecoregions with negative responses of foundation species to MHW variables, and some regions with no response. There is also a neat analysis of the location of the foundation species to the range edge, where the warmer edge showed more impacts for corals and macroalgae.

Will the work be of significance to the field and related fields?

Yes, see below.

How does it compare to the established literature? If the work is not original, please provide relevant references.

While parts of this study have been introduced before (including by some of the authors), the global perspective among multiple coastal foundation species (up to and past the more recent MHW events in the mid to late 2010's) is novel and significant.

Does the work support the conclusions and claims, or is additional evidence needed?

Yes, the conclusions are supported by the results.

Are there any flaws in the data analysis, interpretation and conclusions? Do these prohibit publication or require revision?

I am a little skeptical of the reliance on multiple heatwave variables in the analysis of each region, but overall I am supportive of the analysis as a whole.

Is the methodology sound? Does the work meet the expected standards in your field?

Yes, see above.

Is there enough detail provided in the methods for the work to be reproduced?

This is my main concern with this manuscript. There is not enough information in the methods to delineate what parts of the datasets were used. For example, the dataset in REF 66 contains data from a number of data sources spanning a wide range in the NE Pacific. Also, much of the data in that paper came from the data in REF 65. It is not stated whether those duplicate data were removed from the analysis (or if data from Baja CA from REF 66 were used or not since I believe they are in the same ecoregion as Southern CA. I think that the underlying data for the analyses should be included as a dataset and broken down by foundation species type and ecoregion for reproducibility. Also, more info on the amounts of data from REF 66 should be included in the methods. Also, the n in Suppl Table 1 seems small for the number of PISCO sites in Northern CA.

One other issue I have with the Suppl Table 1 is that the average response of each ecoregion is shown and colored by a loss or gain, yet many do not have significance to the tested variables, so I am not sure how the term 'response' could be used here.

In the Suppl. Table 2 I think that all of the stats vs, MHW variables should be shown for all the ecoregions shown in Suppl Table 1.

Reviewer #3 (Remarks to the Author):

This is a timely, well-written, and interesting paper that provides a global overview of the effects of marine heatwaves (MHWs) on ecologically important foundation species. I suspect it will receive a wide readership. Generally speaking, I was impressed with the authors' efforts. That said, there are a few areas that could be improved.

First, there was not much consideration of the limitations of the data. All such synthesis studies have to contend with things like publication bias, targeted sampling that prioritizes locations or species where effects are expected to be strongest, etc. The authors don't need to write an entire treatise on this, but some consideration would be appropriate.

More importantly, I found some of the statistical analyses hard to follow, and it wasn't clear if some statements (e.g., line 121) were supported by statistics or were just summaries of visually apparent patterns on the global maps. This link between analysis and interpretation was particularly weak for the 'Increasing marine heatwave intensity and duration' section (see detailed comments below). I would recommend explicitly describing each statistical comparison and which terms were included in the model, ideally in a table or tables, and include information on how non-independence in the data (see below) were accounted for. Also, if there was multiple testing of different MHW characteristics within ecoregions (I don't *think* there was, but it wasn't 100% clear to me), I'd recommend a rethink of where alpha is set. With so many tests producing p-values that are marginal ($p > 0.01$), there is a substantial likelihood of Type 2 error if multiple tests were involved. It looks like ecoregion was include (as a fixed effect? random effect?) as part of the global-level GLMs, but the phrasing on line 109-110 doesn't make it clear what variables were tested separately vs. together. Long story short, the model structure needs to be clarified, and the results of each specific test, including all the independent variables used in those tests, needs to be presented in a way that allows the reader to tell what variables were being modeled together. See addition comments on this below as well.

Finally, there are several potential sources of non-independence in the data: multiple sites from the same study, species within a broader clade, and spatial autocorrelation. The authors should describe how these were handled, or justify why they were ignored.

Minor comments by line number

154: "impacted by" doesn't sound quite right; could change to "related to"

153-157: I think these results all come from a GLM in which all of these terms were included

in a single model, but I had the shadow of a doubt as the description could also apply to a series of univariate tests.

159: "exacerbates the health of foundation species" is also odd as no direction is stated and health is usually considered a good thing - perhaps "exacerbates the negative consequences for foundation species"

160-185, Figure 4 & 5, and associated data: again, I'm pretty sure that these results derive from a single model with many explanatory terms included, but if I'm wrong about that, the fact that the authors tested five different MHW characteristics and the most significant p-values for those comparisons were > 0.01 in four out of six cases (Fig 4) and eight out of 12 cases (Fig 5) suggests that your interpretation of significance for many of these ecoregions wouldn't survive an alpha correction to account for multiple tests. I apologize if I'm just being dense, but clarifying the statistical model will be important as it underpins one of the main conclusions that you are trying to make based on p-values that would be marginal if derived from a series of independent tests.

168-169: "statistically significant relationships were found between biological responses and two or more MHW characteristics (Table 1)" - I couldn't find that information in Table 1. Did you mean Extended Data Table 2?

212: warm range edge species should be warm range edge populations

224: Reference 32 is a link that didn't lead anywhere when I clicked it, but even the "Page not found" doesn't appear to be a peer-reviewed paper. Here is a link to a Gonzalez-Espinosa & Donner 2021 paper that shows that cloud cover can reduce coral bleaching: https://onlinelibrary.wiley.com/doi/full/10.1111/gcb.15676?casa_token=LxNT6jHx6W4AAA%3AoZK2WHWmp7tiCo6_OCIGOhAeSqw8P3qOAKeolXzU92IFjDUvNFxzYqfhXzTKycCh-Zm8sqnKGN0TGA

278: I support separating exposed coral reefs from sheltered coral reefs, by why not go ahead and analyze the exposed coral reef data? Perhaps they won't show strong impacts

from MHWs, but I think that would provide a useful counterpoint and will help inform conservation and management strategies. Even if it only appeared in the supplemental data, I still think it would be useful. Regardless of if / how you incorporate patterns on exposed reefs, I would definitely mention the exposed / sheltered distinction in the main text. That is an important caveat to the presentation of the coral data as it biases the perceived strength of impact upwards for corals, and many (most?) readers will miss it if it is hidden away in the Methods.

364: I'm not clear on how you defined the trailing range edge for a species that spans the equator.

373: Unclear what "weighted by number of sites within a localised area for each ecoregion" means

Figure 1a: "average response" should be defined in the caption, along with what the blue regions represent - are they no data, insufficient data, or something else? Finally, and this may not be practical, it would be nice if the color scheme in Fig 1a matched the color scheme in Fig 1b. This last comment applies to Figure 2 as well.

Figure 2: Unlike Fig 1a, the numbering sequence in Fig 2a doesn't make much geographical sense. Also, the colors used for mild and moderate gain in 2b and 2c are pretty similar on my screen.

Figure 3: Do you need the middle panel with the normal curve? If you decide to keep it, you need to describe what the normal curve represents. Population abundance across the range? If that's the intention, we know that the abundant center model doesn't do a good job of representing a lot of marine species (Rafe Sagarin's work). I'd suggest getting rid of it unless you have a compelling reason to keep it. The bar graphs speak for themselves, although boxplots overlain on the actual data points would be better than bar graphs.

Extended Data Table 2: Where do these AIC values come from, and what are they being used for? I didn't find any reference to them anywhere else in the manuscript.

REVIEWER COMMENTS

Reviewer #1 (Remarks to the Author):

Some previous studies have provided evidence that MHWs are impacting foundation species (including corals, kelps and macroalgae) across different locations and regions (e.g., Mediterranean sea, Caribbean Sea, Indo-Pacific...). However, to my knowledge, the present work by Smith et al. is the first study providing a comprehensive, quantitative analysis of the impact of MHWs on critical foundation species across the globe, as well as evidence on how MHWs characteristics (e.g., average intensity, maximum temperature...) and geographic factors (e.g., location within species ranges) may modulate responses. Specifically, the authors combine time series of empirical ecological data (i.e., changes in abundance for macroalgae and seagrasses, and bleaching or mortality observations for corals) and a MHW framework, to quantitatively examine responses of key, habitat-forming foundation species (macroalgae, seagrass and corals) to MHWs in 778 coastal areas located across 72 marine ecoregions. The authors found compelling evidence that intense, summer MHWs play a significant role in foundation species decline. The authors also found that MHW intensity, absolute temperature and location within a species' range are critical factors mediating impacts. Moreover, foundation species have not been affected in some ecoregions, suggestive of some resilience. Overall, I consider that the study is highly timely, addresses a highly relevant topic that is of global interest and has critical and novel results that not only support the conclusions and claims, but also expand our previous knowledge on the topic. The consequences of the reported global decline in marine foundation species driven by MHWs are undeniable for both ecosystems and human societies. The paper is also very well written, with nice figures and sound statistical methods that will be reproducible upon publication of the data.

Thank you for your positive comments on our research. We would also like to thank the reviewer very much for their careful and detailed read, and constructive comments on our manuscript.

Nevertheless, I have some comments that the authors may need to address and/or clarify, which I attach below:

#Title and abstract

Not totally necessary...but since the paper is only focused on observations shallower than 10 m (i.e., cut-off of 10m depth for all datasets), it may be a good idea to include the word "shallow" in the title and/or in the abstract so the readers can easily realize that the paper focuses on this zone. I say this because the impacts of marine heatwaves at deeper depths may be different in different regions even in coastal areas, depending on different factors such as for instance the presence/absence of thermocline.

Done. We have now added the word 'shallow' as clarification in the abstract.

#Main text

Lines 44-117 and 119-187: I am sure authors were planning to do it in the future, but just in case, it would be great to see the headings for these sections as Introduction and Results, respectively.

Done. Thank you – we have now edited the manuscript in line with Nature Communications formatting instructions.

Lines 122-124, Figures 1-2: Maybe it is just my curiosity, but; would it be possible, by simply looking at the text or at Figure 1, to know which 6 ecoregions are the ones that are not impacted? It seems to me that those regions with little impact (<10% of the population impacted) and those with no bleaching at all (0%) have been pooled together as (0-10% category). For instance, when looking at the Extended Data Table 1, I can see that Banda Sea was not impacted. However, Banda Sea appears as 0-10% yellow color in the map (panel a), or mild impact (panel b). While the legend is up front with it and says 0-10% (which includes 0), I believe it would be better to have a separate category for those non-impacted ecoregions (0%; maybe white colors in the map?), and another one for 1-10% (yellow). Otherwise, it may create the feeling of mild impact in a region that is not impacted at all when looking solely at the map. Also, without doing such separation, it is not possible to know which are those ecoregions that are not impacted and have only one data point, since the only way of detecting the true zero is to go to Extended Data Table 1 to see the average, and the average is not shown for regions with less than 3 points.

Done. This is a great idea we had not considered. We have revised the figure as suggested.

Another point that I believe should be addressed is the legend capture with color codes of Figures 1 and 2, panel a. It says “average response”. However, there are some regions whose values (and colors) are not averages but results of a single data point. Authors are upfront with it mentioning the number of replicates for each region in panels b and c, as well as in the main text and in Extended Data Table 1. However, as it is now, the panel a by itself is not totally correct since it still says average responses. Maybe authors could add asterisks in those ecoregion numbers linked in the map to regions that could not be averaged (such as 25, 26 in figure 1 panel a) and then explain what those asterisks mean in the figure text. Alternatively, authors may prefer to add an asterisk in the head of the legend (average response*) so then they can explain that not all ecoregions of the map are averages, or any other solution...

Done. We have now added an asterisks as suggested – thank you.

Another concern in Figures 1 and 2 is related to how categories have been divided in panel a. If 0-10 and 10-20 are different categories; what would happen if there was an average of exactly 10, or 20? Would they be assigned to the first 0-10 or to the second 10-20 group? Shouldn't the categories be divided such as 0, 1-10, 11-20, 21-30...etc to avoid that issue? It has been done in panels b and c and to me it is clearer...

Done. This is correct – we did not write this clearly. We have now adjusted the category divides as suggested.

Finally, since ‘other’ habitat-forming invertebrates refers here (in Figure 1) to gorgonians, and gorgonians are also corals (octocorals), the header of panel c should not say non-coral habitat forming invertebrates, but something like soft coral habitat-forming invertebrates, or non-hard coral habitat forming invertebrates.

Done. Thank you, we have now simplified to terminology related to the animal foundation species. In short, we are now consistently referring all the bleaching FS data to ‘bleaching of scleractinian hard corals’ (or the shorter/simpler ‘hard corals’) and the MME of the non-coral habitat-forming invertebrates as ‘MME of gorgonian soft corals’ (or the shorter/simpler ‘soft corals’). This change makes our terminology more correct and consistent taxonomically,

functionally and method-wise

Line 129-142 (and Extended Table 1): I believe there is a small issue here. For corals, the values in these lines are in %0-100 (e.g., 44% in Mediterranean), as shown also in Extended Table S1. However, for macrophytes, while values are also %0-100 in the text (e.g., 39.3% loss in North California), the corresponding values are extremely low (0.4 loss in California) in Extended Table 1 (while they were supposed to be average response %0-100 as well). I believe that the data may have mistakenly been written as proportions 0-1 in Extended Table 1 for macrophytes, instead of percentages 0-100 (as it was for corals).

Done. Thank you for identifying this typo. We have now corrected the extended data table.

Line 169: Is the reference of Table 1 actually referring to Extended Table S2?

Done. Yes, you are right. We have now corrected this in the manuscript to make this reference clearer.

Line 176: Since other habitat-forming invertebrates refers to gorgonians, consider to specify it here (and maybe throughout the text) (i.e., gorgonians)

We have replaced 'habitat-forming invertebrates' with 'gorgonian soft corals' throughout the manuscript.

#Discussion

Line 222: It would be great to add here some examples of the regions that did not see bleaching despite registered MHWs.

We have added some examples.

Line 248: The authors mention how analyses such as the one performed in the study can be used to ascertain potential climatic refugia or warming-resilient ecosystems by identifying areas at low risk of events or foundation species groups that exhibit some resistance to MHW. Authors also show in the results how some few ecoregions and systems were not impacted or even gained cover, suggesting some potential resilience. Authors do point at some potential causes (oceanography, cloud cover or indicate a level of thermal acclimation...) of the observed lack of bleaching, but I am maybe missing a little bit more discussion about whether the identified non-impacted regions in this study could (or not) actually be potential climatic refugia...and why authors believe so. Similarly, I miss a little bit more information about why some seagrasses and macroalgae have avoided impacts (Torres Strait and Northern Great Barrier Reef) or even increased cover (Bassian) following sometimes severe MHWs.

Done, thank you for this suggestion. We have added a line to say that identification of these areas offer opportunity for further investigation. We have also added a line to discuss why we sometimes observed no or even positive impacts. Note, that this result is not surprising, but are often expected to occur near the poleward range edge of foundation species where species are limited by cold temperatures. Examples of positive impacts include expansion of kelp beds and seagrass meadows in the Arctic or elevated coral growth in subtropical regions. We have added the following references to support this evidence.

Filbee-Dexter, K., Wernberg, T., Fredriksen, S., Norderhaug, K. M., & Pedersen, M. F. (2019). Arctic kelp forests: Diversity, resilience and future. *Global and planetary change*, 172, 1-14,

Marbà, N., Krause-Jensen, D., Masqué, P., & Duarte, C. M. (2018). Expanding Greenland seagrass meadows contribute new sediment carbon sinks. *Scientific Reports*, 8(1), 14024,

Assis, J., Serrão, E. A., Duarte, C. M., Fragkopoulou, E., & Krause-Jensen, D. (2022). Major expansion of marine forests in a warmer Arctic. *Frontiers in Marine Science*, 9, 850368'.

Yamano, H., Sugihara, K., & Nomura, K. (2011). Rapid poleward range expansion of tropical reef corals in response to rising sea surface temperatures. *Geophysical Research Letters*, 38(4).

#Methods

Line 269: If possible, I would rather see Extended Data Table 3 as a Table in the main manuscript, since it contains information that is highly relevant.

Done.

Line 335: Please, either use the word gorgonian, or specify that other habitat-forming species refer to gorgonians only

Done. See our previous comment about simplified terminology for FS. Yes you are correct – it is simply ‘gorgonian soft corals’ for all the MME analyses.

Line 358-360: but single points also appear in the maps of Figures 1 or 2 as averages.

Done. See above – we have now added an asterisk and identified points that are not averages.

Line 382 (and line 115): I am a little bit confused here. It says: “for all ecoregions with ≥ 10 datapoints we ran additional GLMs to test for the relationships between MHW characteristics and foundation species responses at the ecoregion level.” However, ecoregions with exactly 10 datapoints were excluded in Figure 5 (and Extended Table S2) with only one exception (Lesser Sunda). Similarly, for Figure 4 (and Extended Table S2), only ecoregions with >10 datapoints were used (those with 10 were excluded as well), and Central and Southern Great Barrier Reef (with 15 points) was excluded. It doesn't seem to be consistent; am I missing something?

*Adding to my previous comment, in Line 161, it says that only ecoregions with > 10 points were used, which makes me believe that the \geq symbol used in lines 382 and 115 was wrong. That would explain most of it. However, I still do not see why Lesser Sunda (with 10 points) was the unique 10-point region used for corals, nor why Central and Southern Great Barrier Reef (with 15 points) was excluded for the macrophytes case. Maybe It was explained somewhere and I didn't see it, but please clarify.

Done. We appreciate this was confusing. There were some ecoregions where either a) ≥ 10 datapoints for a given species were not available (e.g. Central and Southern GBR), or b) in the case of coral, ≥ 10 datapoints were not available that were presented in the correct format. Some coral data were presented as a range only (i.e. $<10\%$, 10-50%, 50% + bleaching) and we were unable to

use these in our analyses. We have added further clarification to the methods and included new columns in Extended Data Table 1 to indicate the number of datapoints available in the correct format.

#General comment for Figures 1 and 2

I would prefer a clearer separation between panel a and panels b-c in both figures, since as it is now, the color codes of panels b,c (located just below the map) make the impression of being part of panel a. Other option could be to put the color legend of panels b/c below these panels, instead of in below panel a.

Done. Thank you for this suggestion to clarify the figures. We have now, as suggested, added a box around panels b and c in each figure and moved the legend to the bottom of the figure.

#Extended Data

Extended data table 1:

- It would be nice to see also de SD, minimum and maximum values for each ecoregion

Done.

- According to this table, Bassian and Southern Norway have the same average response (%) = 0.1 gain. However, in Figure 2, panel a, whereas Southern Norway has an average response of 0-10% gain, Bassian region seems to have a color of 10-20% gain. If data in Extended Table 1 were correct for macrophytes (0.1 and 0.1), these values should also match in the map and correspond to the 0-10% interval. However, as mentioned above, I believe that the data may have mistakenly been written as proportions 0-1 in the table S1 (instead of percentages 0-100). Since they have also been rounded, they now look equal (0.1 and 0.1), when they should be 11.8% for the Bassian region, and something below 10% for the Norway one? Please, clarify.

Done. Yes, this is correct. The data for macrophytes had been included previously as proportions. It is now revised.

Dr. Daniel Gómez-Gras

Reviewer #2 (Remarks to the Author):

What are the noteworthy results?

This manuscript assesses the response of coastal foundation species to marine heatwave metrics, including max absolute temperature, mean/max/cumulative intensity, and duration. The novel part of this analysis is that the authors examine corals, seagrasses, and macroalgae in the worlds oceans. They find many ecoregions with negative responses of foundation species to MHW variables, and some regions with no response. There is also a neat analysis of the location of the foundation species to the range edge, where the warmer edge showed more impacts for corals and macroalage.

Will the work be of significance to the field and related fields?

Yes, see below.

How does it compare to the established literature? If the work is not original, please provide relevant references.

While parts of this study have been introduced before (including by some of the authors), the global perspective among multiple coastal foundation species (up to and past the more recent MHW events in the mid to late 2010's) is novel and significant.

Does the work support the conclusions and claims, or is additional evidence needed?

Yes, the conclusions are supported by the results.

Are there any flaws in the data analysis, interpretation and conclusions? Do these prohibit publication or require revision?

I am a little skeptical of the reliance on multiple heatwave variables in the analysis of each region, but overall I am supportive of the analysis as a whole.

Is the methodology sound? Does the work meet the expected standards in your field?

Yes, see above.

Thank you for your overall positive response to our manuscript. We would also like to thank the reviewer for their detailed comments below, that have helped us improve the manuscript through the responses and revision process.

Is there enough detail provided in the methods for the work to be reproduced?

This is my main concern with this manuscript. There is not enough information in the methods to delineate what parts of the datasets were used. For example, the dataset in REF 66 contains data from a number of data sources spanning a wide range in the NE Pacific. Also, much of the data in that paper came from the data in REF 65. It is not stated whether those duplicate data were removed from the analysis (or if data from Baja CA from REF 66 were used or not since I believe they are in the same ecoregion as Southern CA. I think that the underlying data for the analyses should be included as a dataset and broken down by foundation species type and ecoregion for reproducibility. Also, more info on the amounts of data from REF 66 should be included in the methods. Also, the n in Suppl Table 1 seems small for the number of PISCO sites in Northern CA.

Done. It is important to note that all data (macroalgae and otherwise) were cross-checked for duplication and all duplicates were removed. We have now clarified and confirmed this in the methods. As recommended by reviewer 1 we now also include extended data table 3 in the main document and have updated the table to indicate the amount of original data that was available from each source. The reason the 'n' in Suppl Table 1 appears small for Northern CA is that data from sites within 8 km of each other were averaged as a conservative approach to avoid over-inflating replication levels (see lines 104 and 348 of the manuscript). E.g. in Northern CA, 11

observations of *Macrocystis* impacts were taken from a total of 30 locations, with some of these locations averaged because of close proximity (and therefore potential for high spatial autocorrelation). We have added a data file including all of the raw data used in analyses to our submission.

One other issue I have with the Supp Table 1 is that the average response of each ecoregion is shown and colored by a loss or gain, yet many do not have significance to the tested variables, so I am not sure how the term 'response' could be used here.

Done. We have replaced the word 'response' with the word 'change'. We have also added to the figure legend a description of the ecoregions where only a single data point was available (i.e., the change could therefore not be averaged or shown with errors, but had to be taken from this single value).

In the Suppl. Table 2 I think that all of the stats vs, MHW variables should be shown for all the ecoregions shown in Suppl Table 1.

Respectfully, we disagree with this suggestion and do not think this is necessary (it may indeed be misleading). We have run GLMs on all ecoregions where ≥ 10 datapoints are available because we argue this is statistically robust. We do not see additional value in running GLMs on Ecoregions with much smaller datasets. For example, we may find significant results from datasets with poor temporal resolutions (e.g., can easily occur due to spurious effects) which would overinflate our conclusions about MHW impacts.

Reviewer #3 (Remarks to the Author):

This is a timely, well-written, and interesting paper that provides a global overview of the effects of marine heatwaves (MHWs) on ecologically important foundation species. I suspect it will receive a wide readership. Generally speaking, I was impressed with the authors' efforts.

That said, there are a few areas that could be improved.

We thank the reviewer for these very positive comments. We have taken on board the detailed comments below which have helped us improve the manuscript through the responses and revision process.

First, there was not much consideration of the limitations of the data. All such synthesis studies have to contend with things like publication bias, targeted sampling that prioritizes locations or species where effects are expected to be strongest, etc. The authors don't need to write an entire treatise on this, but some consideration would be appropriate.

Done. Thank you for this important suggestion. We have now added some text to the discussion that explicitly acknowledge key data limitations.

More importantly, I found some of the statistical analyses hard to follow, and it wasn't clear if some statements (e.g., line 121) were supported by statistics or were just summaries of visually apparent patterns on the global maps. This link between analysis and interpretation was particularly weak for the 'Increasing marine heatwave intensity and duration' section (see detailed comments below).

Done. We have revised the text throughout the results section to make clearer which results are statistically significant and which results are evaluated from 'non-significant trends' (e.g. a trend line can be negative, but because of high data variability not significant). Hopefully this should now provide greater clarity regarding the robustness and significance of our findings.

I would recommend explicitly describing each statistical comparison and which terms were included in the model, ideally in a table or tables, and include information on how non-independence in the data (see below) were accounted for.

Done. We have now added a table as suggested.

Also, if there was multiple testing of different MHW characteristics within ecoregions (I don't *think* there was, but it wasn't 100% clear to me), I'd recommend a rethink of where alpha is set. With so many tests producing p-values that are marginal ($p > 0.01$), there is a substantial likelihood of Type 2 error if multiple tests were involved. It looks like ecoregion was included (as a fixed effect? Random effect?) as part of the global-level GLMs, but the phrasing on line 109-110 doesn't make it clear what variables were tested separately vs. together. Long story short, the model structure needs to be clarified, and the results of each specific test, including all the independent variables used in those tests, needs to be presented in a way that allows the reader to tell what variables were being modelled together. See additional comments on this below as well.

Done. For each ecoregion a single test was performed for all MHW characteristics. This is now clarified better in the text and in the aforementioned table.

Finally, there are several potential sources of non-independence in the data: multiple sites from the same study, species within a broader clade, and spatial autocorrelation. The authors should describe how these were handled, or justify why they were ignored.

Done. As the reviewer highlights, we identified spatial autocorrelation in our dataset to be a major source of non-independence. We have corrected for this to the best of our ability by averaging the effects from multiple sites in close proximity. We also recognise that multiple sites were from the same study or dataset. However even within a dataset, data for each site was typically collected by different people reducing the potential for non-independence. Although we agree that it is possible that similar species within a broader clade may have similar responses, our dataset shows large variation in responses for a single species across different studies. We have therefore elected to retain the full dataset, to ensure we have the highest power to detect the influence of MHW characteristics. We have added a short section to the methods to highlight these points along with a sentence in the discussion.

Minor comments by line number

154: "impacted by" doesn't sound quite right; could change to "related to"

Done. We have changed the relevant text.

153-157: I think these results all come from a GLM in which all of these terms were included in a single model, but I had the shadow of a doubt as the description could also apply to a series of univariate tests.

Done. Correct, these results all come from a GLM in which all of these terms were included in a single model. We have now clarified this further in the text and, as recommended above, have included a table detailing all terms in each model.

159: "exacerbates the health of foundation species" is also odd as no direction is stated and health is usually considered a good thing - perhaps "exacerbates the negative consequences for foundation species"

Done. We have changed the text to read 'exacerbates negative impacts'.

160-185, Figure 4 & 5, and associated data: again, I'm pretty sure that these results derive from a single model with many explanatory terms included, but if I'm wrong about that, the fact that the authors tested five different MHW characteristics and the most significant p-values for those comparisons were > 0.01 in four out of six cases (Fig 4) and eight out of 12 cases (Fig 5) suggests that your interpretation of significance for many of these ecoregions wouldn't survive an alpha correction to account for multiple tests. I apologize if I'm just being dense, but clarifying the statistical model will be important as it underpins one of the main conclusions that you are trying to make based on p-values that would be marginal if derived from a series of independent tests.

Done. We agree this was not very clear in the text. All MHW characteristics were included in a single model in each instance, so we did not run multiple independent tests. We have now clarified this in the text and, as recommended above, have included a table detailing all terms in each model.

168-169: "statistically significant relationships were found between biological responses and two or more MHW characteristics (Table 1)" - I couldn't find that information in Table 1. Did you mean Extended Data Table 2?

Done. Apologies, yes. We have now edited the manuscript text accordingly.

212: Warm range edge species should be warm range edge populations

Done. We have now revised accordingly.

224: Reference 32 is a link that didn't lead anywhere when I clicked it, but even the "Page not found" doesn't appear to be a peer-reviewed paper. Here is a link to a Gonzalez-Espinosa & Donner 2021 paper that shows that cloud cover can reduce coral bleaching:

https://onlinelibrary.wiley.com/doi/full/10.1111/gcb.15676?casa_token=LxNT6jHx6W4AAAAA%3AoZK2WHWmp7tiCo6_OClGOhAeSqw8P3qOAKeolXzU92lFjDUvNFxzYqfhXzTKycCh-Zm8sqnKGN0TGA

Done. Thank you for providing a link to this relevant paper. We have updated the reference accordingly.

278: I support separating exposed coral reefs from sheltered coral reefs, by why not go ahead and analyze the exposed coral reef data? Perhaps they won't show strong impacts from MHWs, but I think that would provide a useful counterpoint and will help inform conservation and management strategies. Even if it only appeared in the supplemental data, I still think it would be useful. Regardless of if / how you incorporate patterns on exposed reefs, I would definitely mention the exposed / sheltered distinction in the main text. That is an important caveat to the presentation of

the coral data as it biases the perceived strength of impact upwards for corals, and many (most?) readers will miss it if it is hidden away in the Methods.

Done. We thank you for this great suggestion for additional analyses. We have now completed and added the suggested analyses for these exposed reefs. We now incorporate all the coral data together essentially doubling the size of that dataset because for other foundation species, datasets were not separated into 'exposed' and 'sheltered'.

364: I'm not clear on how you defined the trailing range edge for a species that spans the equator.

We took the trailing (and leading) range edge of the species that spans the equator as the equatorward (and poleward) extents, respectively. We now provide a more detailed explanation in the text and have improved the 'old' figure 3 (which is now figure 4) to better support the explanation.

373: Unclear what "weighted by number of sites within a localised area for each ecoregion" means

Done. We have now clarified the sentence to read 'weighted by the number of sites that were averaged within a geographically similar localised area, as described above'. Hopefully this is now clearer (we agree that the phrase 'For each ecoregion' was confusing and has therefore been deleted).

Figure 1a: "average response" should be defined in the caption, along with what the blue regions represent – are they no data, insufficient data, or something else?

Done. We now better describe what an 'average response' is (see above), and have included wording to show that the blue coloured ecoregions represent ecoregions where no data were available.

Finally, and this may not be practical, it would be nice if the color scheme in Fig 1a matched the color scheme in Fig 1b. This last comment applies to Figure 2 as well.

Done. Following the suggestion from reviewer 1 we have now clarified the figure, by separating out the legends to make it clearer which legends are relevant to which panels in the plot. However, we have kept the colour schemes as in our originally submitted manuscript. Some of our data were only available as ranges (e.g. <10% bleaching, 10-50% bleaching, >50% bleaching). Therefore for b and c we chose to keep the colour scheme as ranges to enable us to retain all of our data. Comparatively, where we averaged the data on the global maps of ecoregions we feel that the higher level categories gives a better representation of average responses by ecoregion.

Figure 2: Unlike Fig 1a, the numbering sequence in Fig 2a doesn't make much geographical sense.

Done. Please note that the numbers follow a logical sequence across the map from left to right showing first results for seagrass and then for macroalgae (and to follow the patterns of data in panels b and c).

Also, the colors used for mild and moderate gain in 2b and 2c are pretty similar on my screen.

Done. We have updated the colours for clarity.

Figure 3: Do you need the middle panel with the normal curve? If you decide to keep it, you need to

describe what the normal curve represents. Population abundance across the range? If that's the intention, we know that the abundant center model doesn't do a good job of representing a lot of marine species (Rafe Sagarin's work). I'd suggest getting rid of it unless you have a compelling reason to keep it. The bar graphs speak for themselves, although boxplots overlain on the actual data points would be better than bar graphs

Done. We have edited the middle panel to show illustrative performance curves for species from the northern hemisphere, equatorial region, and southern hemisphere, indicating where species are more likely (warm range areas) or less likely (cool range areas) to experience negative impacts of MHWs. This addition is based on this reviewer's previous comments and to support the explanation above on how a point in a species range was assessed when that species spanned the equator.

Extended Data Table 2: Where do these AIC values come from, and what are they being used for? I didn't find any reference to them anywhere else in the manuscript.

Done. The AIC values were used to guide model choice. We have now provided some additional text to the methods to explain this.

REVIEWERS' COMMENTS

Reviewer #1 (Remarks to the Author):

I have read through the revised version of the manuscript by Smith et al. and I find their responses and changes satisfactory. I was happy to see that they carefully addressed each of the comments and raised concerns. I also appreciate the effort made to further clarify methodology while including exposed reefs in the analyses, as suggested by other reviewers. I am delighted to recommend acceptance of the manuscript.

Some last few suggestions that authors might find convenient:

-It might just be personal preference, but now that the figures 2 and 3 are clearer with panels b and c being perfectly separated from panel a, I would rather see the severity color legends (i.e., proportion of organisms affected and change in cover/density) at the top of the block b-c, right before both (b) titles. That is, where you had it before. It would work great now because panel a is now effectively separated from b-c.

-Since ecoregions with 1-2 data points (e.g., Hawaii, Northern Galapagos...) are being represented in the map, I believe they also should be included in Extended Table 1. You could use NA for the range and SD columns, and place the single value in the average column, using an asterisk to indicate that those are not averages but single values (like you did in the figures). Indeed, if for the map you averaged ecoregions with 2 observations, you could also add this average in Extended Table 1, even if specifying that averages from less than 3 points were not considered for further analyses. Whatever the case, I believe including information on ecoregions with 1 or 2 observations in Extended Table 1 could be a good idea.

-Lastly, throughout the manuscript, I would suggest not using the term "soft corals" alone to refer to gorgonians, since true soft corals (lack an axis; e.g., *Xenia* sp.) are totally different from gorgonians (present an axis and are semi-rigid; e.g., *Gorgonia* sp.). I see that using gorgonian soft corals can be convenient to reinforce the fact that gorgonians are softer than scleractinians, but I would avoid using the term "soft coral" alone, unless the sentence is

referring to all non-stony corals as a group (i.e., including as well true soft corals), such as it may happen sometimes in the intro or discussion sections.

Dr. D. Gómez-Gras

Reviewer #2 (Remarks to the Author):

Thank you for your comprehensive comments and revisions. I am fine with recommending the manuscript for publication.

Reviewer #3 (Remarks to the Author):

The authors have done a good job of responding to the reviewers' comments, and the issues that I had with the previous draft have largely been clarified. I only had two remaining suggestions, both minor in nature.

First, two of the main conclusions may be influenced in part by the decision to restrict the analysis to heatwaves that occur in the summer for extratropical locations: 1) effects were predominantly negative, and 2) trailing edge populations were disproportionately impacted. There may be important impacts at other times of year, and those may or may not mirror the conclusions drawn from summer data. There are also other variables like reproduction and recruitment which are more difficult to measure than MMEs and bleaching and thus less amenable to this type of data synthesis. Using the chosen response variables and restricting analyses to summer MHWs is not a flaw per se - no manuscript can cover everything - but I would recommend adding a caveat in the discussion that points out that we currently lack a good understanding of how certain biological responses are changing, and how MHWs outside of the summer season may drive ecological change.

Second, in Figure 2a, is there a way to indicate that "change" is always negative for these taxa, e.g., by mentioning on line 675 that the color codes represent no change or varying degrees of increased bleaching or mortality?

REVIEWERS' COMMENTS

Reviewer #1 (Remarks to the Author):

I have read through the revised version of the manuscript by Smith et al. and I find their responses and changes satisfactory. I was happy to see that they carefully addressed each of the comments and raised concerns. I also appreciate the effort made to further clarify methodology while including exposed reefs in the analyses, as suggested by other reviewers. I am delighted to recommend acceptance of the manuscript.

Thank you. We thoroughly appreciate the time you have taken to consider our manuscript and your positive comments.

Some last few suggestions that authors might find convenient:

-It might just be personal preference, but now that the figures 2 and 3 are clearer with panels b and c being perfectly separated from panel a, I would rather see the severity color legends (i.e., proportion of organisms affected and change in cover/density) at the top of the block b-c, right before both (b) titles. That is, where you had it before. It would work great now because panel a is now effectively separated from b-c.

Thank you for this suggestion. We have now moved the legend back to the top of panels b and c.

-Since ecoregions with 1-2 data points (e.g., Hawaii, Northern Galapagos...) are being represented in the map, I believe they also should be included in Extended Table 1. You could use NA for the range and SD columns, and place the single value in the average column, using an asterisk to indicate that those are not averages but single values (like you did in the figures). Indeed, if for the map you averaged ecoregions with 2 observations, you could also add this average in Extended Table 1, even if specifying that averages from less than 3 points were not considered for further analyses. Whatever the case, I believe including information on ecoregions with 1 or 2 observations in Extended Table 1 could be a good idea.

Thank you for this useful suggestion. We have added the relevant data to Extended Table 1.

-Lastly, throughout the manuscript, I would suggest not using the term "soft corals" alone to refer to gorgonians, since true soft corals (lack an axis; e.g., *Xenia* sp.) are totally different from gorgonians (present an axis and are semi-rigid; e.g., *Gorgonia* sp.). I see that using gorgonian soft corals can be convenient to reinforce the fact that gorgonians are softer than scleractinians, but I would avoid using the term "soft coral" alone, unless the sentence is referring to all non-stony corals as a group (i.e., including as well true soft corals), such as it may happen sometimes in the intro or discussion sections.

Thank you for this suggestion. We have now changed 'soft corals' to 'gorgonian soft corals' throughout the manuscript.

Dr. D. Gómez-Gras

Reviewer #2 (Remarks to the Author):

Thank you for your comprehensive comments and revisions. I am fine with recommending the manuscript for publication.

Thank you. We really appreciate the time you have taken to consider our manuscript.

Reviewer #3 (Remarks to the Author):

The authors have done a good job of responding to the reviewers' comments, and the issues that I had with the previous draft have largely been clarified. I only had two remaining suggestions, both minor in nature.

Thank you for your positive comments and the time you have taken to reconsider the manuscript.

First, two of the main conclusions may be influenced in part by the decision to restrict the analysis to heatwaves that occur in the summer for extratropical locations: 1) effects were predominantly negative, and 2) trailing edge populations were disproportionately impacted. There may be important impacts at other times of year, and those may or may not mirror the conclusions drawn from summer data. There are also other variables like reproduction and recruitment which are more difficult to measure than MMEs and bleaching and thus less amenable to this type of data synthesis. Using the chosen response variables and restricting analyses to summer MHWs is not a flaw per se - no manuscript can cover everything - but I would recommend adding a caveat in the discussion that points out that we currently lack a good understanding of how certain biological responses are changing, and how MHWs outside of the summer season may drive ecological change.

The reviewer makes an excellent point, with which we fully agree. Given the importance of temperature in influencing all biological processes, MHWs occurring outside of summer will likely elicit different responses, some of which may be positive. We have added a caveat to the discussion to clarify this.

Second, in Figure 2a, is there a way to indicate that "change" is always negative for these taxa, e.g., by mentioning on line 675 that the color codes represent no change or varying degrees of increased bleaching or mortality?

Thank you for this helpful suggestion. We have now added a sentence at the recommended point.